# PRMT3-mediated arginine methylation of IGF2BP1 promotes oxaliplatin resistance in liver cancer

Yunxing Shi[1,2,4], Yi Niu[1,2,4], Yichuan Yuan[1,2,4], Kai Li[1,2], Chengrui Zhong[1,2], Zhiyu Qiu[1,2], Keren Li[1,2], Zhu Lin[1,2], Zhiwen Yang[1], Dinglan Zuo[1], Jiliang Qiu[1,2], Wei He[1,2], Chenwei Wang[1,2], Yadi Liao[1], Guocan Wang [3] ✉, Yunfei Yuan [1,2] ✉ & Binkui Li [1,2] ✉

Although oxaliplatin-based chemotherapy has been effective in the treatment of hepatocellular carcinoma (HCC), primary or acquired resistance to oxaliplatin remains a major challenge in the clinic. Through functional screening using CRISPR/Cas9 activation library, transcriptomic profiling of clinical samples, and functional validation in vitro and in vivo, we identify PRMT3 as a key driver of oxaliplatin resistance. Mechanistically, PRMT3-mediated oxaliplatin-resistance is in part dependent on the methylation of IGF2BP1 at R452, which is critical for the function of IGF2BP1 in stabilizing the mRNA of HEG1, an effector of PRMT3-IGF2BP1 axis. Also, PRMT3 overexpression may serve as a biomarker for oxaliplatin resistance in HCC patients. Collectively, our study defines the PRTM3-IGF2BP1-HEG1 axis as important regulators and therapeutic targets in oxaliplatin-resistance and suggests the potential to use PRMT3 expression level in pretreatment biopsy as a biomarker for oxaliplatin-resistance in HCC patients.

Hepatocellular carcinoma (HCC) is the most common major type of primary liver cancer and the third leading cause of cancer-related death worldwide[1]. Chemotherapy, delivered via trans-arterial chemoembolization (TACE), which led to hypoxia, or hepatic arterial infusion (HAIC), which does not lead to hypoxia, has been used as a major therapeutic approach in the treatment of HCC. Among the various chemotherapeutic agents, oxaliplatin (OXA)-based HAIC has emerged as a promising approach and significantly improves the clinical outcome of a subset of patients[2-4]. However, most HCC patients do not respond to OXA-based HAIC due to primary resistance[3]. Also, the benefits in patients who respond to OXA-based HAIC are short-lived due to the development of acquired resistance[3-5]. Thus, a better understanding of the mechanisms underlying the resistance to OXA-based chemotherapy may improve the clinical outcome of HCC patients.

Multiple biological processes, such as metabolism, apoptosis, hypoxia, DNA damage repair, and epigenetic modification, are involved in OXA resistance[6-9]. Although the resistance mechanisms associated with OXA have been explored in several cancers, the molecular mechanisms of OXA resistance in HCC remain elusive. Protein arginine methylation has been implicated in multiple biological processes, including DNA damage response, RNA processing, and gene expression[10]. Accumulating evidence indicates that regulators of protein arginine methylation are involved in various diseases, including cancer. Thus, targeting protein arginine methyltransferases (PRMTs) represents a very promising therapeutic strategy for delaying tumor progression and overcoming therapeutic resistance[11,12]. Yet, the role of PRMTs in OXA resistance in HCC needs to be further explored. Protein arginine methyltransferase 3 (PRMT3), a member of the PRMTs

[1]State Key Laboratory of Oncology in South China and Collaborative Innovation Center for Cancer Medicine, Guangzhou, China. [2]Department of Liver Surgery, Sun Yat-Sen University Cancer Center, Guangzhou, China. [3]Department of Genitourinary Medical Oncology, The University of Texas MD Anderson Cancer Center, Houston, TX, USA. [4]These authors contributed equally: Yunxing Shi, Yi Niu, Yichuan Yuan. ✉e-mail: gwang6@mdanderson.org; yuanyf@mail.sysu.edu.cn; libk@sysucc.org.cn

family, mediates the asymmetric di-methylation of arginine using S-adenosyl-L-methionine (SAM) as a donor. Unlike other members of the PRMT family, PRMT3 contains a C2H2 zinc finger domain which is critical for substrate recognition[13] and is mainly located in the cytoplasm rather than distributed in the cytoplasm and nucleus[14]. Previous studies have shown that the substrates of PRMT3 include RPS2, HMGA1a, and HMGA1b[15,16]. In pancreatic cancer, PRMT3 has been shown to methylate GAPDH to regulate glucose metabolism[17] and promote gemcitabine resistance by upregulating the expression of ABCA1[18]. These studies suggest that PRMT3 may play an important role in the tumor progression and therapeutic resistance.

In this present study, we harness the power of genome-wide CRISPR activation (CRISPRa)-based functional genomics[19] and unbiased transcriptomic profiling of clinical samples to systematically uncover drivers of OXA resistance in HCC. We functionally validate one of the top candidates, PRMT3, as a driver of OXA resistance in HCC. Also, we leverage orthogonal proteomic, transcriptomic, and epitranscriptomic approaches to elucidate the molecular mechanism underlying the role of PRMT3 in OXA resistance. We demonstrate that IGF2BP1 is a key substrate of PRMT3 and its methylation at arginine 452 is critical for PRMT3-mediated OXA resistance. Furthermore, we show that IGF2BP1 promotes OXA resistance by stabilizing its target transcript *HEG1* in an N6-methyladenosine (m6A)-dependent manner. Moreover, we demonstrate the clinical relevance of PRMT3 overexpression in HCC, as its overexpression is strongly associated with poor clinical outcomes and poor responses to OXA treatment. Thus, our study defines the PRMT3-IGF2BP1-HEG1 axis in driving OXA resistance in HCC.

## Results

### CRISPRa screen and transcriptome analysis of patient samples identify PRMT3 as a candidate driver for OXA resistance in HCC

To identify drivers involved in OXA resistance, we integrated a functional genomics approach via CRISPRa screen with transcriptomic profiling of clinical samples that are responsive or non-responsive to OXA treatment (Supplementary Fig. 1a, b). We performed CRISPRa screen in HepG2 cells, a hepatoblastoma-derived cell line[20] that is highly sensitive to OXA as compared to HCC cell lines Huh7 and PLC-8024 (IC50: HepG2 4.76 μM; Huh7: 7.89 μM; PLC-8024: 18.19 μM) (Supplementary Fig. 1c). Using a CRISPRa library that contains 70,290 unique sgRNA sequences targeting 23,430 human genes, HepG2 cell overexpressing dCas9 protein infected with lentiviral sgRNAs were treated with vehicle or 2 μM OXA for 7 days. We chose 2 μM OXA based on the IC50 study and the effects of various concentrations of OXA (0.5, 1, 2, 4 μM) on cell proliferation and cell death of HepG2 cells during the 7 days of treatment. We found that 2 μM OXA treatment for 7 days significantly inhibited proliferation (95%) and induced cell death of HepG2 cells (Supplementary Fig. 1d), which would provide a strong selection pressure to uncover drivers of OXA resistance in the screening. HepG2 cells treated with OXA and vehicle were subjected to next-generation sequencing of genomic DNA (gDNA) to identify genes that are negatively and positively associated with OXA resistance. We then used Model-based Analysis of Genome-wide CRISPR/Cas9 Knockout (MAGeCK) to identify hits from our CRISPR screening as described[21]. The quality control measurements indicated that sequencing reads had reasonable base qualities (>25) and the percentage of mapped reads (Supplementary Fig. 1e, f). Moreover, the distributions of normalized gRNA read counts in two groups were comparable (Supplementary Fig. 1g). Using a cutoff of $|\log_2 FC| \geq 1$, we identified 761 genes that are positively associated with OXA resistance and 286 genes that are negatively associated with OXA resistance (Fig. 1a; Supplementary Data. 1). To identify candidate genes that are clinically relevant to OXA resistance, we performed RNA-seq of needle biopsy samples from treated patients with advanced HCC who were defined as responders and non-responders based on mRECIST 1.1

criteria after at least 4 cycles of OXA-based chemotherapy delivered via HAIC (see Supplementary Materials and Methods for details). Differential gene expression analysis identified 3428 upregulated and 3483 downregulated genes in the non-responsive tumor samples compared to the responsive samples (Fig. 1b; Supplementary Data. 2). Consistent with the aggressive nature of the non-responsive tumors, Gene Set Enrichment Analysis (GSEA) showed that non-responsive tumors were enriched for the expression of genes involved in epithelial-mesenchymal-transition (EMT), IL6-JAK-STAT3, KRAS, and angiogenesis, which were among the top 10 activated pathways (Supplementary Data. 3). To narrow down the candidate genes for functional validation, we focused on genes upregulated in OXA-resistant HCC using a stringent cutoff ($\log_2 FC \geq 4$, a total of 942 genes) and the top 100 hits in the CRISPRa screen, which resulted in seven common genes (Fig. 1c). Among these 7 genes, *PRMT3*, an arginine methyltransferase ranked as the 6th hit in the CRISPRa screen, drew our attention, because of the critical role of protein arginine methylation in tumor progression and therapeutic resistance[22,23]. We observed significant enrichment of all three sgRNAs for *PRMT3* in OXA-treated HepG2 cells (Fig. 1d, Supplementary Data. 4). Also, we confirmed that the three *PRMT3*-specific sgRNAs, together with sgRNAs specific for multiple genes used in a previous study[24], efficiently activated their corresponding target genes when transfected into HepG2 cells (Supplementary Fig. 1h). Since we found that *PRMT3* mRNA was upregulated by 4.35-fold in HCCs from OXA-based HAIC non-responders (Fig. 1e), we examined whether *PRMT3* expression was affected by OXA treatment in HCC cells. Interestingly, PRMT3 mRNA and protein were upregulated in PLC-8024 and Huh7 cells treated with OXA (Fig. 1f; Supplementary Fig. 1i), suggesting a role for PRMT3 in the adaptive response of HCC to OXA treatment. To examine whether PRMT3 is overexpressed in OXA-resistant HCC cell lines, we generated OXA-resistant PLC-8024 and Huh7 cells (PLC-8024-R & Huh7-R) by subjecting these cells to OXA treatment for 6 months. We then compared the IC50 of OXA in the parental lines and the OXA-resistant sublines. We found that OXA-resistant sublines (PLC-8024-R and Huh7-R) have a much higher IC50 (fold changeå 5) than their parental counterparts (Fig. 1g; Supplementary Fig. 1j). Also, OXA-induced apoptosis was dramatically reduced in PLC-8024-R & Huh7-R compared to their parental controls (Fig. 1h; Supplementary Fig. 1k). Importantly, PRMT3 expression is upregulated at the mRNA and protein levels in both PLC-8024-R and Huh7-R cells compared to their parental counterparts (Fig.1i, j). Collectively, our data suggest that PRMT3 overexpression may render HCC cells resistant to OXA treatment.

### PRMT3 promotes OXA resistance in vitro and in vivo

To determine the role of PRMT3 in OXA resistance, we first examined the effect of PRMT3 overexpression (OE) on the response of HepG2 cells to OXA (Fig. 2a). We found that PRMT3 OE increased the IC50 of OXA in HepG2 cells compared to the vector control (Fig. 2b). Also, PRMT3 OE enhanced the growth of HepG2 cells in the presence of OXA (Fig. 2c). Interestingly, PRMT3 OE enhanced the growth of HepG2 cells in the absence of OXA (Fig. 2c), suggesting a role for PRMT3 in HCC progression. We then verified the effects of SGC707, a PRMT3-specific inhibitor[25], on the response of HCC cells to OXA treatment. We found that the induction of PRMT3 expression by OXA is comparable in both the control and SGC707-treated cells (Supplementary Fig. 2a), consistent with previous reports that SGC707 inhibits PRMT3 enzymatic activities and has no effect on its protein stability[25]. Strikingly, SGC707 treatment completely abolished the effect of PRMT3 OE on the growth of HepG2 cells (Fig. 2c; Supplementary Fig. 2b). Moreover, PRMT3 OE significantly reduced OXA-induced apoptosis compared to the vector control, and PRMT3 inhibition by SGC707 restored the sensitivity of PRMT3-OE HepG2 cells to OXA treatment (Fig. 2d, e).

We then examined the effect of PRMT3 KO/KD on the response of PLC-8024, Huh7, and their OXA-R sublines to OXA treatment, since

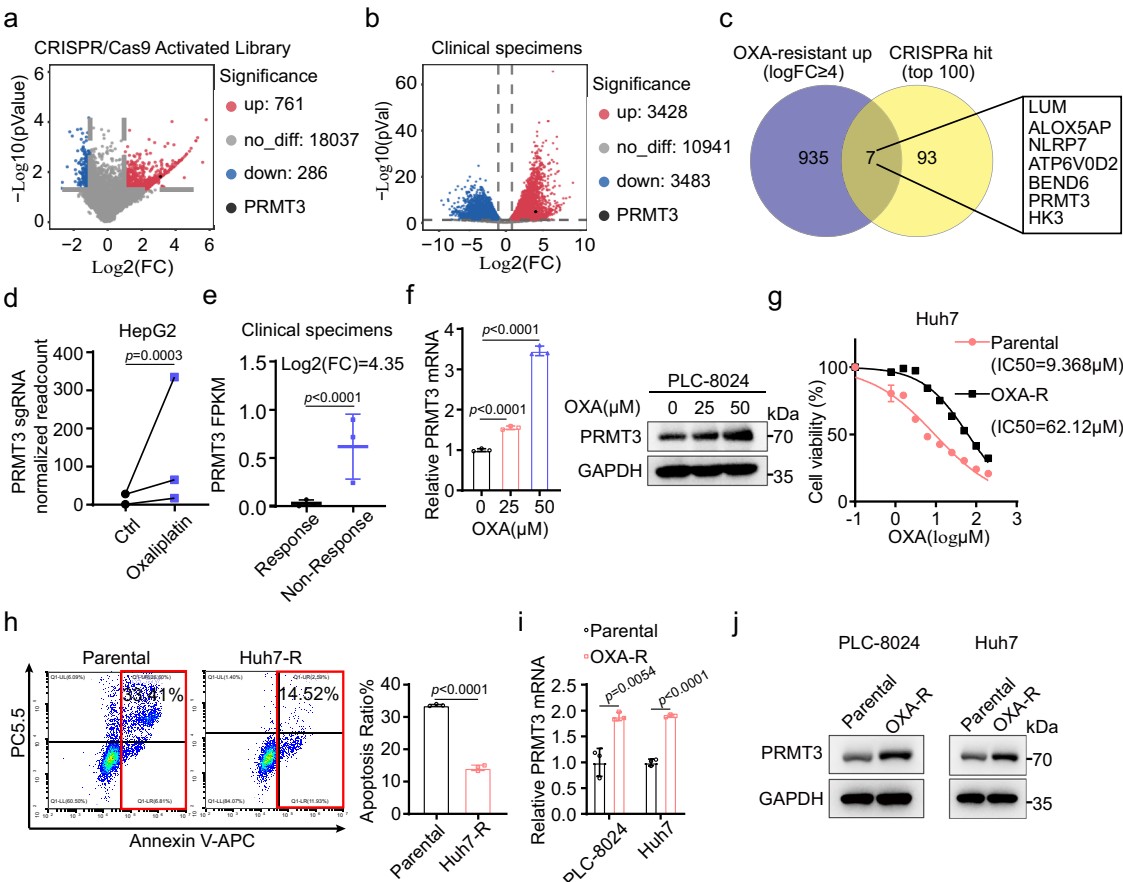

**Fig. 1 | CRISPR activation screen and transcriptome analysis of patient samples identify PRMT3 as a candidate driver for OXA resistance in HCC. a** Volcano plot reveals differential gRNAs targeting genes of genome-wide CRISPRa screen during OXA treatment. **b** Volcano plot shows the differentially expressed genes identified from RNA-seq analysis of HCC patient samples treated with OXA-based HAIC (Responders VS Non-responders). **c** Venn diagram showing top candidate genes involved in OXA resistance based on CRISPRa screen and RNA-seq analysis of HCC patient samples. **d** *PRMT3* sgRNAs were enriched in OXA-treated HepG2 cells compared to vehicle-treated HepG2 cells. **e** Comparison of *PRMT3* mRNA expression (FKPM) in tumors from patients treated with OXA who were defined as responders and non-responders. **f** The mRNA and protein level of PRMT3 in PLC-8024 cells treated with OXA (0, 25, and 50 μM) for 48 h. **g** The IC50 of OXA in Huh7 parental and Huh7-R cells. **h** Apoptosis of Huh7 parental and Huh7-R cells treated with OXA (40 μM) using flow cytometry analysis of Annexin V staining. **i** The mRNA expression of *PRMT3* in OXA-resistant PLC-8024 and Huh7 cells and the corresponding parental cells. **j** WB analysis of PRMT3 expression in OXA-resistant PLC-8024 and Huh7 cells and the corresponding parental cells. For **e**, **f**, **h**, **i**, *n* = 3 biologically independent samples. For **g**, *n* = 5 biologically independent samples. For western blot assay in **f** and **j**, *n* = 3 independent experiments. Data in **e–i** are presented as mean ± SD. Data were analyzed by two-sided Student's t test in **f**, **h** and **I**, NbiomWald Test in **b** and **e**, and learned mean-variance model in **a** and **d**. Source data are provided as a Source Data file.

they are more resistant to OXA compared to HepG2 cells (Fig. 1g; Supplementary Fig. 1c, g). The efficiency of PRMT3 KO by two independent sgRNAs in the pooled cells was confirmed by Western blot, qRT-PCR, and genomic sequencing (Fig. 2f; Supplementary Fig. 2c–e). The efficiency of PRMT3 KD by two independent siRNAs was confirmed by Western blot (Fig. 2f). We found that PRMT3 KO/KD reduced the IC50 of OXA in PLC-8024, Huh7 and the OXA-R sublines (Fig. 2g; Supplementary Fig. 2f). Also, PRMT3 KO/KD in PLC-8024, Huh7, PLC-8024-R, and Huh7-R cells potentiated OXA-mediated growth suppression as shown by colony formation assay and CCK8 viability assay (Fig. 2h; Supplementary Fig. 2g–i). Furthermore, PRMT3 KO/KD in PLC-8024, Huh7, and Huh7-R cells sensitize these cells to OXA-induced apoptosis compared to WT cells as shown by FACS analysis of Annexin V staining (Fig. 2i; Supplementary Fig. 2j). Moreover, PRMT3 inhibitor SGC707 treatment enhanced the OXA-induced growth suppression (Fig. 2j) and apoptosis (Fig. 2k; Supplementary Fig. 2k) in PLC-8024, Huh7, and Huh7-R cells. Interestingly, PRMT3 also impaired the proliferation of PLC-8024 cells in the absence of OXA treatment (Supplementary Fig. 2l), suggesting a role for PRMT3 in HCC progression. To examine the effect of PRMT3 KO on the response of HCC cells to OXA in vivo, we subjected tumor-bearing mice implanted with

PRMT3-KO and -WT PLC-8024 cells to vehicle or OXA treatment (Supplementary Fig. 3a). We found that PRMT3 KO dramatically sensitized PLC-8024 cells to OXA treatment, as shown by reduced tumor sizes and weights (Fig. 2l–n). Similarly, SGC707 treatment improved the response of tumor-bearing mice to OXA treatment (Fig. 2o–q). Decreased expression of Ki67 and increased cleaved caspase 3 were observed in the tumor tissues from the PRMT3-KO cells treated with OXA and tumors from PRMT3-WT cells treated with SGC707 + OXA compared to their corresponding control (Supplementary Fig. 3b, c). Taken together, these results indicate that PRMT3 promotes HCC cell proliferation and survival and contributes to OXA resistance in vitro and in vivo.

## PRMT3 methylates IGF2BP1 at R452

Given that PRMT3 functions through its arginine methyltransferase activity, we performed mass spectrometry (LC/LC-MS) analysis of proteins co-precipitated with PRMT3 in PLC-8024 and HepG2 cells to identify potential PRMT3 substrates (Supplementary Fig. 4a). We identified 84 PRMT3-interacting proteins that were overlapped between these two cell lines (Fig. 3a; Supplementary Data. 5). Interestingly, IGF2BP1, a member of the IGF-2 mRNA-binding proteins

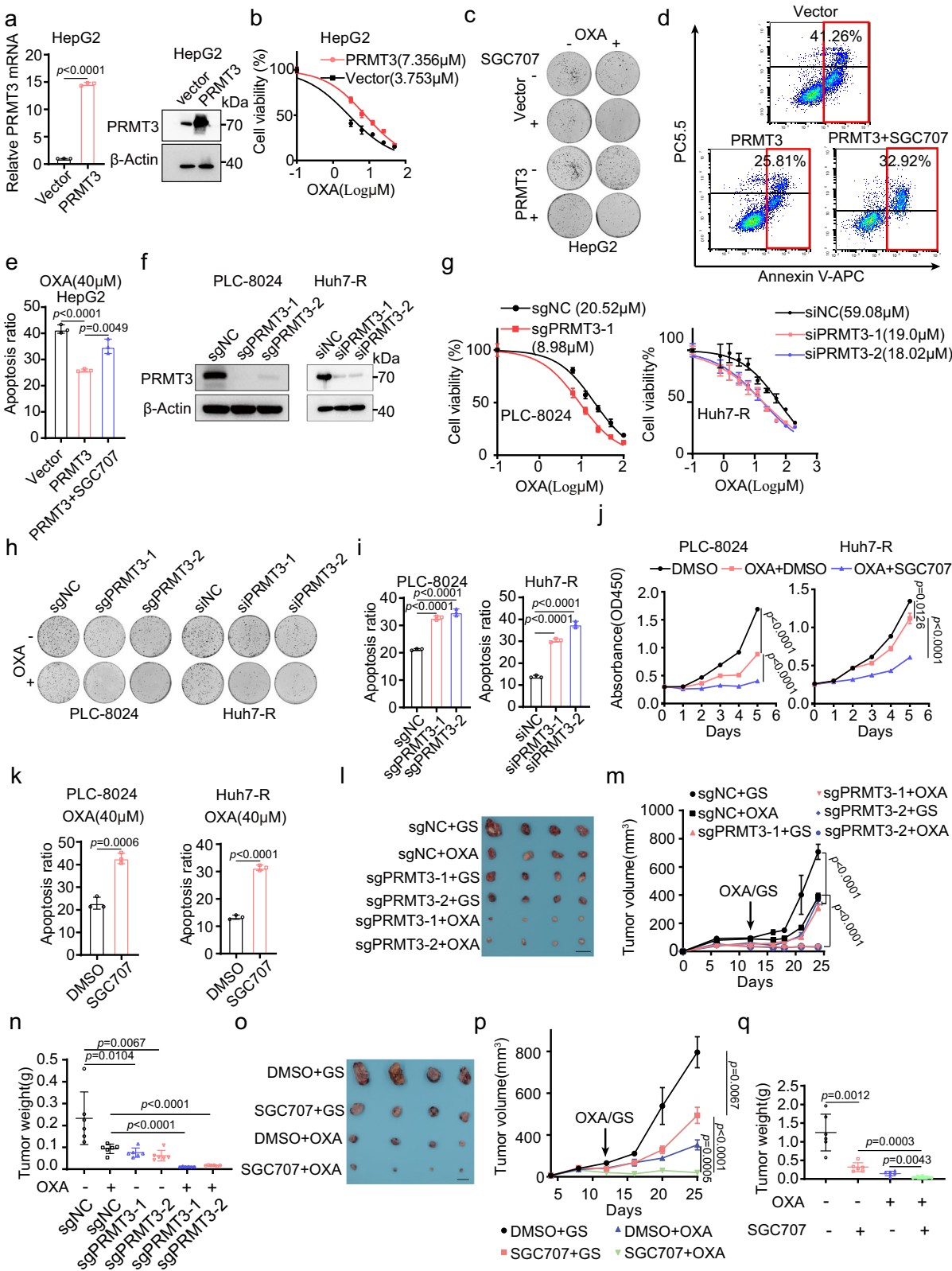

(IGFB2BPs) that play an oncogenic role in multiple cancer types[26], was among the top 6 most abundantly pulled down proteins by PRMT3 (Fig. 3b; Supplementary Fig. 4a). To confirm our IP-MS findings, we examined whether IGF2BP1 interacted with PRMT3 at the endogenous level. Indeed, we found that endogenous PRMT3 efficiently pulled down endogenous IGF2BP1 in PLC-8024 and Huh7-R cells (Fig. 3c, d). Similarly, endogenous IGF2BP1 efficiently pulled down PRMT3 in PLC-

8024 and Huh7 cells (Fig. 3c, d). Also, immunofluorescence (IF) staining showed that PRMT3 and IGF2BP1 colocalize in the cytoplasm in PLC-8024, Huh7, Huh7-R, and HCC tissue from patients (Fig. 3e; Supplementary Fig. 4b). We then examined the effect of PRMT3 OE on arginine methylation of IGF2BP1. We found that PRMT3 OE increased the asymmetric dimethylarginine (ADMA) of IGF2BP1 in HepG2 cells (Fig. 3f). Also, PRMT3 KD or PRMT3 inhibitor SGC707 significantly

**Fig. 2 | PRMT3 promotes OXA resistance in vitro and in vivo. a** PRMT3 expression as shown by qRT-PCR and Western blot analysis in PRMT3 OE cells and control cells. **b** The IC50 of OXA in PRMT3 OE and control HepG2 cells. **c–e** The effects of PRMT3 OE and SGC707 on cell growth (0.5 μM OXA) as shown by colony formation, and on apoptosis (40 μM OXA) by flow cytometry analysis. **f** Western blot analysis of PRMT3 expression in PRMT3 KO/KD and control cells. **g** The IC50 of OXA in PRMT3 KO/KD and control cells. **h** The effect of PRMT3 KO/KD on cell proliferation (0.5 μM OXA) as shown by colony formation. **i** The effect of PRMT3 KO/KD on apoptosis (40 μM OXA) as shown by flow cytometry analysis. **j** CCK8 assay to measure the effects of SGC707 (100 μM) on cell proliferation (1 μM OXA). **k** The effect of SGC707 (100 μM) on apoptosis (40 μM OXA) in HCC cells. **l** The effect of PRMT3 KO on the tumor growth of subcutaneously implanted PLC-8024 cells compared to control cells treated with OXA (5 mg/kg) or vehicle control in nude mice ($n = 6$). Scale bars, 1 cm. The measurement of tumor volumes (**m**) and tumor weights (**n**) of PRMT3 KO PLC-8024 cells and control cells treated with OXA (5 mg/kg) or vehicle control ($n = 6$). **o** The effect of SGC707 (20 mg/kg) on the tumor growth of subQ implanted PLC-8024 cells treated with OXA (5 mg/kg) or vehicle control in nude mice (n = 6). Scale bars, 1 cm. The measurement of tumor volumes (**p**) and tumor weights (**q**) of subQ implanted PLC-8024 cells treated with vehicle, OXA (5 mg/kg), SGC707 (20 mg/kg), and OXA + SGC707 ($n = 6$). For **a**, **c–e**, **g**, and **i–k**, $n = 3$ biologically independent samples. For **b**, $n = 3$ biologically independent samples. For western blot assay in **a** and **f**, n = 3 independent experiments. Data in **a**, **b**, **e**, **g**, **i**, **j**, **k**, **m**, **n**, **p**, and **q** are presented as mean ± SD. Data were analyzed by two-sided Student's *t* test in **a**, **e**, **i**, **j**, **k**, **n**, and **q** and one-way ANOVA adjusted for multiple comparisons for **m**, **p**. Source data are provided as a Source Data file.

reduced the ADMA of IGF2BP1 in PRMT3-OE HepG2, PLC-8024, Huh7 and Huh7-R cells (Fig. 3f, g; Supplementary Fig. 4c), suggesting IGF2BP1 is a substrate of PRMT3. Importantly, our mass spectrometry analysis showed that R452 was the only methylation site of IGF2BP1 (Supplementary Fig. 4d; Supplementary Data. 6). The IGF2BP1 amino acid sequence surrounding R452 was evolutionarily conserved across multiple species (Fig. 3h) and matched the consensus methylation motif, indicating the methylation of this residue may have important biological significance. To determine whether PRMT3 regulates the methylation of R452, we generated R452 to lysine (K) mutant of FLAG-tagged IGF2BP1 (R452K mutant) (Fig. 3i). We overexpressed FLAG-IGF2BP1 or FLAG-IGF2BP1-R452K mutant in HEK293T cells and found that ADMA was abolished in FLAG-IGF2BP1-R452K mutant but not in the WT IGF2PB1 R452K (Fig. 3j). Thus, our data demonstrated that PRMT3 methylated IGF2BP1 at R452.

### R452 methylation of IGF2BP1 is required for OXA resistance

Since we found that PRMT3 plays an important role in OXA resistance, we speculated that arginine methylation of IGF2BP1 might contribute to PRMT3-mediated OXA resistance. To this end, we knocked down IGF2BP1 in PRMT3-OE HCC cells (Supplementary Fig. 5a, b) and examined its effect on the response of HCC cells to OXA. We found that IGF2BP1 KD significantly diminished the effect of PRMT3-OE on HepG2 cell proliferation in the presence of OXA (Fig. 4a). IGF2BP1 KD also abolished the anti-apoptotic effect of PRMT3 OE in HepG2 cells in the presence of OXA (Fig. 4b; Supplementary Fig. 5c). Furthermore, IGF2BP1 KD in PRMT3-OE HepG2 cells reduced the IC50 of OXA compared to PRMT3-OE cells transfected with control siRNA (Supplementary Fig. 5d). IGF2BP1 KD also sensitized PLC-8024 and Huh7 cells to OXA-induced growth suppression and apoptosis (Fig. 4c; Supplementary Fig. 5e–h). Additionally, IGF2BP1 KD sensitized Huh7-R cells to OXA-induced apoptosis (Fig. 4c). However, IGF2BP1 OE in PRMT3-KO cells did not reverse the OXA-induced growth suppression and apoptosis (Fig. 4d, e; Supplementary Fig. 5i).

To examine the role of R452 methylation in regulating IGF2BP1 function, we compared the effect of overexpressing IGF2BP1-WT and IGF2BP1-R452K mutant on the proliferation and OXA-induced apoptosis in IGF2BP1-KD cells. The expression of IGF2BP1 WT almost fully rescued the defect in cell proliferation caused by IGF2BP1-KD and reduced OXA-induced apoptosis in PLC-8024 and Huh7-R cells (Fig. 4f–h; Supplementary Fig. 5j). On the contrary, R452K mutation significantly diminished the ability of IGF2BP1 in promoting cell proliferation and suppressing OXA-induced apoptosis in PLC-8024 and Huh7-R cells (Fig. 4f–h; Supplementary Fig. 5j). Since R452 is the sole arginine residue that was methylated by PRMT3, we examined whether IGF2BP1 function depends on PRMT3-mediated arginine methylation. We examined the effect of overexpressing IGF2BP1-R452K mutant and IGF2BP1-WT on the response of PRMT3-KO PLC-8024 cells to OXA treatment. We found that both IGF2BP1-WT and R452K mutant had no significant difference in cell proliferation in the PRMT3-KO cells in the

presence or absence of OXA (Fig. 4i). However, the co-expression of PRMT3 with IGF2BP1-WT in PRMT3-KO cells enhanced cell proliferation in the presence of OXA treatment (Fig. 4i). On the contrary, co-expression of PRMT3 with IGF2BP1-R452K mutant had a less profound effect on the cell proliferation than co-expression of PRMT3 with IGF2BP1-WT (Fig. 4i). Similarly, there was no significant difference between IGF2BP1-WT OE and IGF2BP1-R452K mutant OE on OXA-induced apoptosis (Fig. 4j; Supplementary Fig. 5k). The co-expression of PRMT3 with R452K mutant is less effective in antagonizing OXA-induced apoptosis compared to the co-expression of PRMT3 with IGF2BP1-WT (Fig. 4j; Supplementary Fig. 5k). Interestingly, PRMT3-OE still enhanced cell proliferation and reduced apoptosis in IGF2BP1-R452K mutant-expressing cells (Fig. 4i, j; Supplementary Fig. 5k), suggesting other PRMT3 substrates may contribute to the effects of PRMT3 OE. To determine the significance of R452 methylation of IGF2BP1 in OXA resistance in vivo, we examined the impact of IGF2BP1-R452K mutation and the co-expression of PRMT3 on the growth of PRMT3-KO PLC-8024 cells in the subcutaneous xenograft model treated with OXA. We found that PRMT3-OE but not IGF2BP1 OE rescued the growth defect observed in PRMT3-KO cells (Fig. 4k–m). Also, IGF2BP1-R452K mutant significantly diminished the effect of PRMT3-OE on tumor growth (Fig. 4k–m). These findings are consistent with our in vitro findings. Collectively, our data suggest that R452 methylation of IGF2BP1 is in part required for PRMT3-mediated OXA resistance.

### PRMT3 and IGF2BP1 regulate HEG1 expression in an N[6]-methyladenosine (m6A)-dependent manner

Since IGF2BP1 was reported to act as a reader for m6A, the most prevalent modification in eukaryotic RNAs, and promote the stability of m6A-modified transcripts[27,28], we sought to identify its downstream effectors that mediate OXA resistance in HCC by using MeRIP (m6A)-sequencing and RIP sequencing. Our MeRIP (m6A)-seq on PLC-8024 cells identified 14162 genes whose RNAs contained an increased m6A modification compared to the input (fold enrichment cutoff 1.2) (Supplementary Fig. 6a–c; Supplementary Data. 7). We identified an m6A consensus sequence motif that is similar to previous reports[28] (Supplementary Fig. 6a). The m6A modifications are mainly localized in CDS, 3′ UTR, and stop codon (Supplementary Fig. 6b) of the coding genes (Supplementary Fig. 6c). Gene ontology (GO) analysis showed that genes with m6A modification were enriched for the mRNA processing, gene expression, DNA damage response, and DNA repair pathways, which were similar to the deregulated pathways identified in PRMT3 KD cells compared to control cells (Fig. 5a; Supplementary Data. 8). Next, we performed RIP-seq in PLC-8024 cells and identified 1481 genes as IGF2BP1 target transcripts (Supplementary Data. 9). Because IGF2BP1 controls mRNA stability, we performed RNA-seq of IGF2BP1-KD PLC-8024 cells and control cells to identify genes that are modulated by IGF2BP1, with a focus on genes that are downregulated upon IGF2BP1 KD (Supplementary Data. 10). By examining m6A-containing transcripts, IGF2BP1 target transcripts, and downregulated

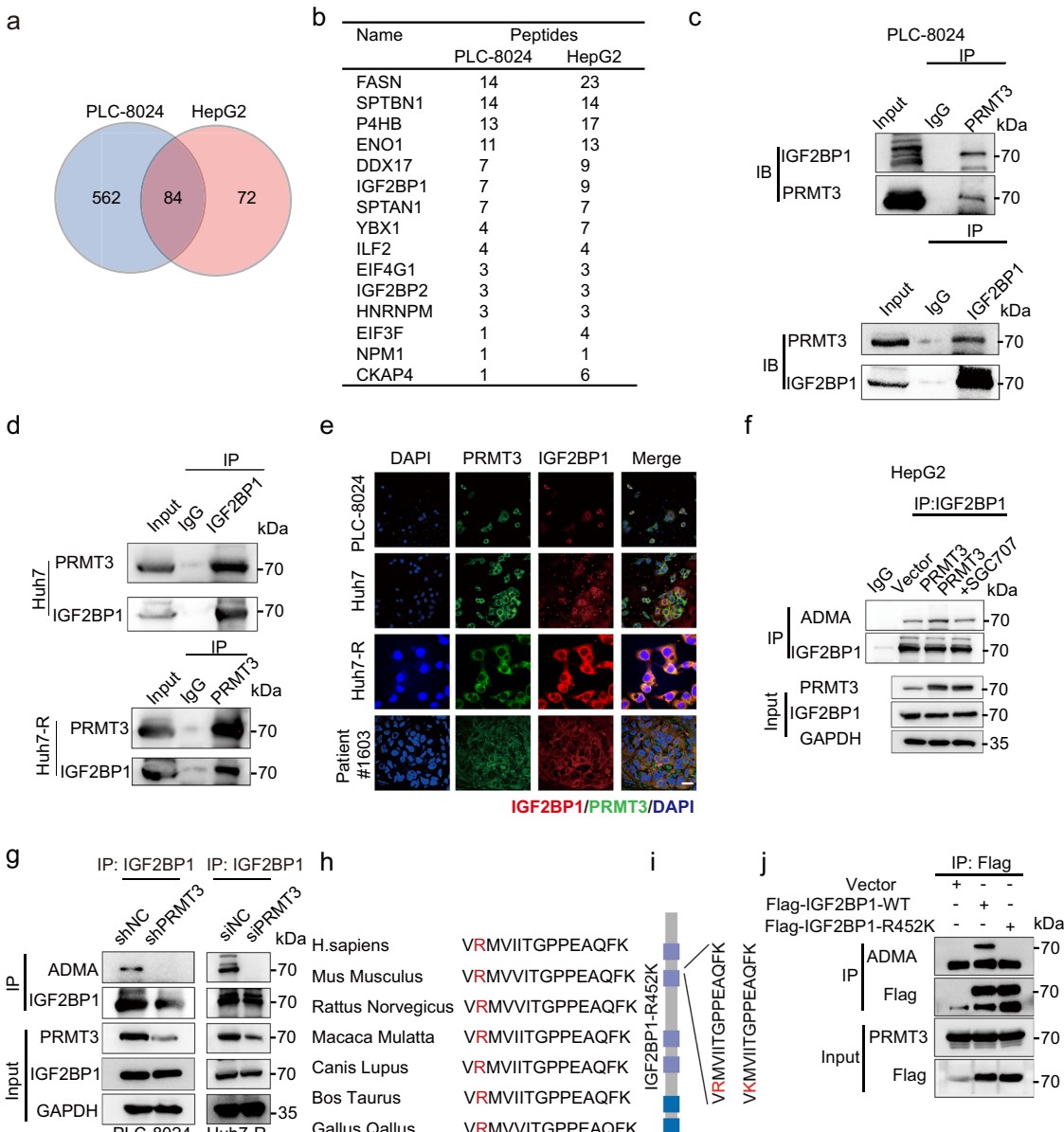

**Fig. 3 | PRMT3 methylates IGF2BP1 at R452. a** Venn Diagram showing the 84 common proteins pulled down by PRMT3 in PLC-8024 and HepG2 cells which were analyzed by liquid chromatography/tandem mass spectrometry (LC-MS/MS). **b** The number of peptide fragments (95% CI) of proteins pulled down by PRMT3 in PLC-8024 cells and HepG2 cells identified from the IP-MS analysis. **c** WB analysis showed that endogenous PRMT3 and IGF2BP1 interact with each other in PLC-8024 cells using reciprocal co-immunoprecipitation. **d** WB analysis showed that endogenous IGF2BP1 pulled down endogenous PRMT3 in Huh7 cells, and endogenous PRMT3 pulled down endogenous IGF2BP1 in Huh7-R cells. **e** Immunofluorescence staining showed the co-localization of PRMT3 (green) and IGF2BP1 (red) in HCC cells and HCC patient samples. Scale bar, 50 μm. **f** WB analysis of immunoprecipitated

IGF2BP1 to determine the effect of PRMT3 OE and PRMT3 inhibition by SGC707 on arginine methylation of IGF2BP1 in PRMT3-KO PLC-8024 cells using asymmetric dimethylarginine antibody (ADMA). **g** WB analysis of immunoprecipitated IGF2BP1 to determine the effect of PRMT3 KD on arginine methylation of IGF2BP1 in PLC-8024/Huh7-R cells. **h** The sequences surrounding R452 of IGF2BP1 are evolutionarily conserved across multiple species. **i** A scheme showing the sequences of Flag-tagged IGF2BP1-WT and IGF2BP1-R452K mutant. **j** WB analysis of immunoprecipitated Flag-tagged IGF2BP1-WT and IGF2BP1-R452K mutant showed that R452K mutation dramatically reduced ADMA signal in HEK293 cells overexpressing Flag-tagged IGF2BP1-WT and IGF2BP1-R452K mutant. For **c–g**, and **j** $n = 3$ independent experiments. Source data are provided as a Source Data file.

genes in PRMT3 KD cells and IGF2BP1-KD cells, we identified BTB Domain Containing 7 (BTBD7) and heart development protein with EGF-like domains 1 (*HEG1*) as candidate genes that are regulated by IGF2BP1 (Fig. 5b). Because HEG1, but not BTBD7, was reported to directly contribute to survival, metastasis, and chemoresistance in HCC through regulation of Wnt/beta-catenin signaling[29,30], a major signaling pathway regulating liver homeostasis and tumorigenesis[31,32], we decided to examine the role of HEG1 in PRMT3- mediated OXA resistance.

As expected, *HEG1* mRNA contains m6A modification and binding to IGF2BP1 (Fig. 5c). Also, we found that PRMT3 or IGF2BP1 positively regulated HEG1 expression: PRMT3 KO/KD or IGF2BP1 KD downregulated *HEG1* mRNA and protein expression levels in PLC-8024, Huh7, and Huh7-R cells (Fig. 5d–g; Supplementary Fig. 6d, e). PRMT3 OE upregulated *HEG1* mRNA and protein in HepG2 cells (Fig. 5h, i). Furthermore, the upregulation of HEG1 in PRMT3-OE HepG2 cells was reversed upon IGF2BP1 KD (Fig. 5i), suggesting that the regulation of HEG1 by IGF2BP1 is dependent on PRMT3-mediated

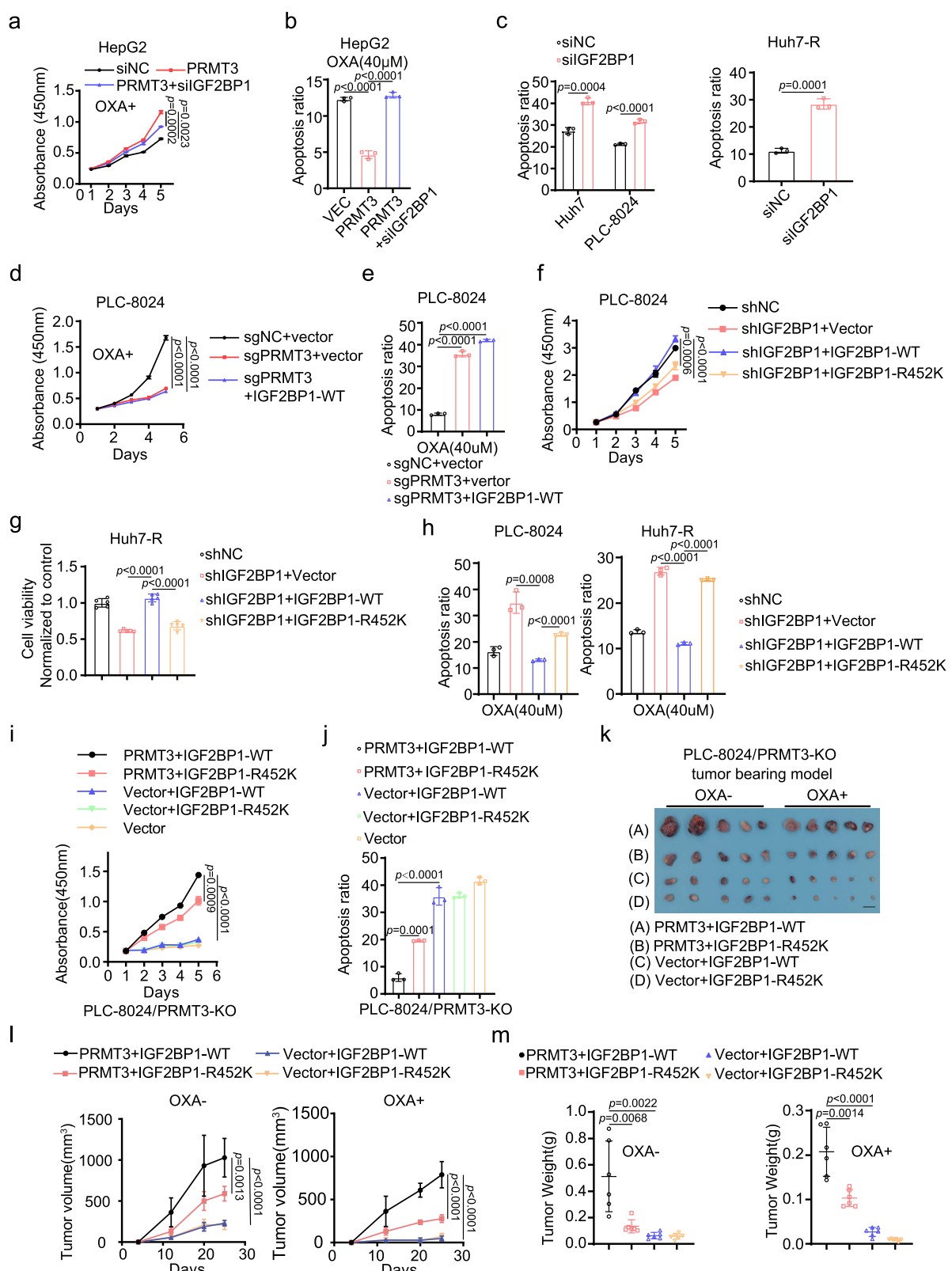

arginine methylation. To test this, we overexpressed PRMT3 in PRMT3-KO cells expressing Flag-tagged IGF2BP1-WT and IGF2BP1-R452K mutant. We found that HEG1 expression was increased upon PRMT3 OE in cells expressing IGF2BP1-WT but not IGF2BP1-R452 mutant (Fig. 5j-k). This confirmed that PRMT3-mediated arginine methylation of IGF2BP1 plays an important role in the regulation of HEG1 expression. Since IGF2BP1 positively regulates RNA stability

(m6A regulator)[28], we examined the stability of *HEG1* mRNA in IGF2BP1 KD cells. As expected, IGF2BP1 KD significantly reduced the half-life of *HEG1* mRNA in PLC-8024 and Huh7-R cells (Fig. 5l; Supplementary Fig. 6f). PRMT3 KO also reduced the stability of *HEG1* mRNA (Fig. 5m; Supplementary Fig. 6g). Conversely, PRMT3 OE prolonged the half-life of *HEG1* mRNA, which was reversed by treatment with PRMT3 inhibitor SGC707 or IGF2BP1 KD in PRMT3-OE cells

**Fig. 4 | R452 methylation of IGF2BP1 is required for OXA resistance. a, b** The effect of IGF2BP1 KD on the proliferation (1 μM OXA) and apoptosis (40 μM OXA) of PRMT3-OE HepG2 cells. **c** The effect of IGF2BP1 KD on apoptosis (40 μM OXA) of PLC-8024, Huh7, and Huh7-R cells. **d** The effect of IGF2BP1-WT OE on cell proliferation (1 μM OXA) of PRMT3-KO PLC-8024 cells. **e** The effect of IGF2BP1-WT OE on apoptosis (40 μM OXA) of PRMT3-KO PLC-8024 cells. **f** The effect of IGF2BP1-WT and IGF2BP1-R452K mutant OE on cell proliferation of IGF2BP1-KD PLC-8024 cells and control cells treated with OXA (1 μM). **g** The effect of IGF2BP1-WT and IGF2BP1-R452K mutant OE on cell proliferation (1 μM OXA) of IGF2BP1-KD Huh7-R cells and control cells. **h** The effect of IGF2BP1-WT and IGF2BP1-R452K mutant OE on apoptosis (40 μM OXA) of IGF2BP1-KD PLC-8024/Huh7-R cells and control cells. **i, j** The effect of IGF2BP1-WT and R452K mutant OE on the cell proliferation (1 μM OXA) and apoptosis (40 μM OXA) of PRMT3-KO PLC-8024 cells and PRMT3-KO PLC-8024 cells with PRMT3 OE. **k** The IGF2BP1-WT and R452K mutant OE on the

growth of PRMT3-KO PLC-8024 cells and PRMT3-KO PLC-8024 cells with PRMT3 OE, which were subcutaneously implanted in nude mice (n = 6), in the presence and absence of OXA treatment (5 mg/kg OXA). Scale bars, 1 cm. **l** The measurement of tumor volumes to determine the effect of IGF2PB1 WT and R452K mutant OE on the growth of PRMT3-KO PLC-8024 cells and PRMT3-KO PLC-8024 cells with PRMT3 OE, which were treated with OXA or vehicle (n = 6). **m** The tumor weights of PRMT3-KO PLC-8024 cells and PRMT3-KO PLC-8024 cells with PRMT3 OE which co-expressed IGF2BP1-WT or IGF2BP1-R452K at the endpoint of the experiment (Day 25) (n = 6). For **a–e** and **h–j** n = 3 biologically independent samples. For **f** and **g**, n = 5 biologically independent samples. Data in **a–j**, **l** and **m** are presented as mean ± SD. Data were analyzed by two-sided Student's t test in **a–j** and **m**, and one-way ANOVA adjusted for multiple comparisons for **l**. Source data are provided as a Source Data file.

(Fig. 5n-o). Restoring PRMT3 prolonged the half-life of *HEG1* mRNA in IGF2BP1-WT expressing cells but not in IGF2BP1-R452K expressing cells (Fig. 5p). We then examined whether arginine methylation of IGF2BP1 affects its binding to HEG1 mRNA using RIP assay. We found that IGF2BP1-R452K, a mutant defective in arginine methylation, impaired the binding of IGF2BP1 to *HEG1* (Fig. 5q). Additionally, treatment of HCC cells with 3-deazaadenosine (DAA), which inhibits SAH hydrolase and interrupts insertion of m6A into mRNA substrates[33,34], significantly decreased total m6A in PLC-8024 and Huh7 cells (Supplementary Fig. 6h) and the expression of *HEG1* mRNA and protein in the parental and Huh7-R HCC cells (Fig. 5r; Supplementary Fig. 6i, j), suggesting that m6A is critical for the stability of HEG1 mRNA. DAA treatment also downregulated HEG1 expression induced by PRMT3 OE (Supplementary Fig. 6k, l). These results revealed that *HEG1* mRNA stability was regulated by PRMT3-mediated arginine methylation of IGF2BP1 in an m6A-dependent manner.

## The effect of PRMT3 and IGF2BP1 on OXA resistance is dependent on HEG1

Since HEG1 is regulated by PRMT3, we determined whether HEG1 was the downstream effector of PRMT3 in the regulation of OXA resistance. We found that HEG1 KD (Fig. 6a) abolished the effect of PRMT3 OE on HepG2 cells in the presence of OXA treatment, resulting in decreased cell proliferation and decreased colony formation ability (Fig. 6b–e). HEG1 KD similarly reversed the effects of PRMT3 OE on cell proliferation and colony formation in the absence of OXA treatment (Fig. 6b–e). Furthermore, HEG1 KD re-sensitized PRMT3-OE HepG2 cells to OXA-induced apoptosis (Fig. 6f; Supplementary Fig. 7a). We also over-expressed HEG1 in PRMT3 KD Huh7-R cells (Fig. 6g). As expected, HEG1 OE rescued the effects of PRMT3 KD on cell proliferation with and without OXA treatment and OXA-induced apoptosis (Fig. 6h, i). Also, HEG1 KD in PLC-8024 and Huh7 cells (Supplementary Fig. 7b) led to reduced colony formation (Supplementary Fig. 7c), reduced cell proliferation (Supplementary Fig. 7d), and increased OXA-induced apoptosis (Supplementary Fig. 7e). HEG1 KD also sensitized Huh7-R cells to OXA-induced apoptosis (Supplementary Fig. 7f). Moreover, knockdown of HEG1 markedly overcame the OXA resistance induced by PRMT3 OE in HepG2 cells in vivo as shown by weekly measurement of tumor volumes and tumor weights at the endpoint (Fig. 6j, k; Supplementary Fig. 7g). HEG1 KD also suppressed PRMT3-induced HCC growth in the absence of OXA treatment in vivo (Fig. 6j, k; Supplementary Fig. 7g). To examine the effect of HEG1 KD on cancer cell proliferation in vivo, we performed Ki67 IHC staining in treated tumors. We found that HEG1 KD in PRMT3-OE HepG2 cells dramatically reduced the number of Ki67+ cells and increased the number of cleaved caspase 3+ cells compared to PRMT3-OE HepG2 cells in the presence and absence of OXA treatment (Fig. 6l, m). Collectively, these results showed that HEG1 plays a critical role in the PMMT3-mediated OXA resistance in HCC cells.

## High PRMT3 expression correlates with poor clinical outcomes and poor therapeutic responses to OXA-based HAIC in HCC patients

Since we found that PRMT3 plays an important role in OXA resistance, we first determined whether high PRMT3 expression, as determined by IHC staining, correlates with poor response to OXA-based HAIC in 36 cases of post-treatment surgical HCC samples. Strong and moderate IHC staining intensity of PRMT3 was defined as PRMT3-high whereas weak and negative staining was defined as PRMT3-low (Supplementary Fig. 8a, b). The responses of the patients were defined by Response Evaluation Criteria in Solid Tumors (RECIST; ver. 1.1) and modified RECIST (mRECIST) criteria. The mRECIST criteria followed the recommendation posed by the original RECIST publication to encourage amendments for tumors presenting unique complexities and for the evaluation of anti-cancer therapies. It was proposed as a way of adapting the RECIST criteria to the particularities of HCC[35]. We found that patients with a better response to OXA-based HAIC, as shown by MRI imaging, had lower PRMT3 expression than patients with higher PRMT3 expression (Fig. 7a, b). Based on the RECIST criteria, 11 of 18 patients (61.1%) experienced objective response in the PRMT3-low group while only 27.8% in the PRMT3-high group experienced objective response (P < 0.05) (Fig. 7c–e). A total of 11 patients experienced a partial response and no one suffered from disease progression in the PRMT3-low group. However, two patients suffered from disease progression and only five experienced a partial response in the PRMT3-high group. We observed similar findings when mRECIST is used for evaluating the patient responses (Fig. 7c–e). We then explored whether PRMT3 expression levels could be used as a potential biomarker for response to OXA-based HAIC, we generated the receiver operating characteristic (ROC) curves and calculated the area under the curve (AUC), which reflect the sensitivity and specificity of a biomarker, using the post-treatment dataset. We found that the AUCs were 0.67 and 0.70 based on RECIST and mRECIST criteria, respectively (Fig. 7f), suggesting that PRMT3 could potentially serve as a biomarker for responses to OXA-based HAIC.

These findings prompted us to investigate whether PRMT3 expression levels in pretreatment HCC biopsies could correlate with the response to OXA-based HAIC. We enrolled 31 patients diagnosed with advanced HCC at the Sun Yat-sen University Cancer Center from July 1, 2020, to June 30, 2021. Pretreatment biopsy samples were collected from this cohort of patients prior to OXA-based chemotherapy. Based on PRMT3 IHC staining intensity, all the enrolled patients were divided into PRMT3-high or PRMT3-low groups according to IHC score (Fig. 7g). A total of 24 patients had elevated alpha-fetoprotein (AFP) at the study baseline. Only 1 (5.9%) patient experienced an increase in AFP from baseline in the PRMT3-low group while 5 of 14 (35.7%) experienced elevated AFP in the PRMT3-high group (Fig. 7h). Based on the RECIST v1.1 and mRECIST, the number of patients experiencing PD (progressive disease) was significantly lower among PRMT3-Low patients (5.9%) compared with PRMT3-High patients (57.1%, P = 0.006). Patients in the PRMT3-low group had significantly higher

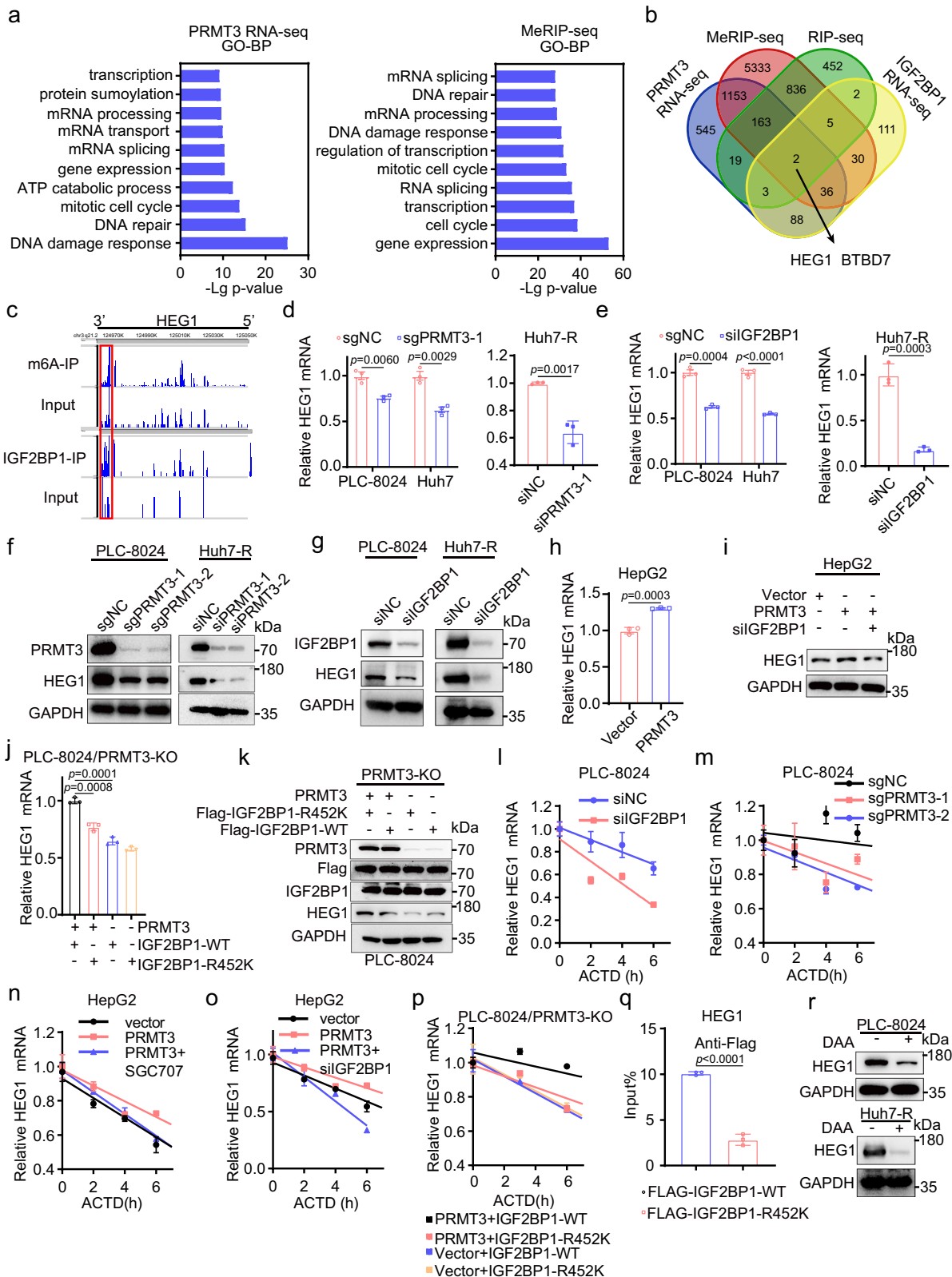

ORR (objective response rate) compared with the PRMT3-high group (RECIST $P = 0.031$, mRECIST $P = 0.005$) (Fig. 7i, j). Furthermore, the PRMT3 expression level has an AUC of 0.72 and 0.732 for predicting objective response based on the RECIST and mRECIST criteria (Fig. 7k). These data strongly suggest that PRMT3 expression levels in pretreatment biopsy samples could also serve as a potential biomarker to stratify patients for OXA-based HAIC treatment.

Patient-derived cell (PDC) models have emerged as a powerful tool for preclinical studies that recapitulate the patients' response to therapy[36]. Thus, we decided to compare the response to OXA using HCC PDCs derived from patients with PRMT3-high or PRMT3-low tumors as determined by IHC presurgical tumor specimen. We generated two HCC PDC lines (S1: PRMT3-high, and S2: PRMT3-low) (Fig. 7l). We found that PDC S1 line was more resistant to OXA treatment (high

**Fig. 5 | PRMT3 and IGF2BP1 regulate HEG1 expression in an m6A-dependent manner. a** The top 10 gene ontology (GO) terms for DEGs from RNA-seq analysis of PRMT3 KD and control cells and for genes containing m6A modification from MeRIP-m6A-seq. **b** Venn Diagram identified *HEG1* and *BTBD7* as the two candidate genes regulated by IGF2BP1 by integrating lists of genes downregulated in PRMT3 KD cells, genes downregulated in IGF2BP1 KD cells, transcripts directly bound by IGF2BP1, and transcripts containing m6A modifications. **c** IGV plots showing examples of shared and specific peaks of *HEG1* from MeRIP-seq and RIP-seq. Peaks are represented as subtracted read densities (IP and input). **d, e** *HEG1* mRNA expression in PRMT3-KO/ IGF2BP1-KD and control cells (PLC-8024 and Huh7-R) as shown by qRT-PCR. **f, g** WB analysis of HEG1 in PRMT3-KO/ IGF2BP1-KD and control cells (PLC-8024 and Huh7-R). **h** *HEG1* mRNA expression in PRMT3-OE HepG2 cells and control cells as shown by qRT-PCR. **i** HEG1 protein expression in control, PRMT3-OE, and PRTM3-OE + IGF2BP1 KD HepG2 cells. *HEG1* mRNA (**j**) /protein (**k**) levels in PRMT3-KO PLC-8024 cells infected with vector + IGF2BP1-WT, vector + IGF2BP1-R452K mutant, PRMT3 + IGF2BP1-WT, PRMT3 + IGF2BP1-R452K mutant. **l, m** The effect of IGF2BP1-KD/ PRMT3-KO on the stability of *HEG1* mRNA. **n** The effect of PRMT3 OE and PRMT3 OE + SGC707 on the mRNA stability of *HEG1*. **o** The mRNA stability of HEG1 in PRMT3-overexpressing HepG2 cells transfected with the IGF2BP1 siRNA or corresponding control. **p** The mRNA stability of *HEG1* in PRMT3-KO PLC-8024 cells infected with vector + IGF2BP1-WT, vector + IGF2BP1-R452K mutant, PRMT3 + IGF2BP1-WT, PRMT3 + IGF2BP1-R452K mutant. **q** The direct binding of IGF2BP1-WT and IGF2BP1-R452K with *HEG1* transcript as determined by RIP assay. **r** HEG1 protein expression in PLC-8024 and Huh7-R cells treated with 50 μM 3-deazaadenosine (DAA) or vehicle. For **d, e, h, j** and **l–q**, *n* = 3 biologically independent samples. For **f, g, i, k** and **r**, *n* = 3 independent experiments. Data in **d, e, h, j, l–q** are presented as mean ± SD. Data were analyzed by two-sided Student's *t* test in **d, e, h, j** and **q**. Source data are provided as a Source Data file.

IC50) compared to PDC S2 line (Fig. 7m), which is consistent with our findings in patients that high PRMT3 is associated with poor response to OXA (Fig. 7n, o). Since we demonstrated that PRMT3 inhibitor sensitized HCC cell lines to OXA treatment, we examined the effect of PRMT3 inhibitor SGC707 on the responses of HCC PDCs to OXA. We found that SGC707 sensitized both cell lines from patients to OXA treatment (Fig. 7p), with PRMT3-high S1 cells (from patient 1) displaying a more profound decrease in cell viability than PRMT3-low S2 cells (from patient 2) upon SGC707 treatment (Fig. 7p). Collectively, our data strongly suggest that PRMT3 may serve as a biomarker for predicting the response of HCC patients to OXA-based chemotherapy.

## Discussion

OXA-based HAIC has been recently used as an effective maintenance therapy in patients with advanced HCC[4,5]. However, de novo resistance to OXA-based HAIC (or chemotherapy) has been frequently observed in HCC patients, as about 54% of HCC patients had little response to OXA-based treatment[3]. Also, for those patients who respond to OXA-based treatment, acquired resistance could eventually develop. However, the underlying molecular mechanisms that confer OXA resistance in HCC cells remain largely unknown. Our study identified the PRMT3-IGF2BP1-HEG1 axis as a regulator of OXA resistance in HCC. Also, our study suggests that targeting PRMT3 may be an effective approach to improve the response of HCC cells to OXA treatment. Furthermore, PRMT3 expression in biopsy specimens may predict patients' response to OXA-based HAIC.

Protein arginine methyltransferases have emerged as attractive therapeutic targets in cancer. PRMTs regulate a diverse array of biological processes, including transcription, splicing, RNA biology, DNA damage response, and cell metabolism[37–40]. A better understanding of the mechanisms by which how these enzymes drive tumor progression and therapeutic resistance has provided the rationale for targeting them in oncology. Previous studies had revealed the role of PRMT3 in chemoresistance, tumor growth, and metabolism[17,18]. However, the driver mediating OXA resistance remains poorly understood. Using functional screening with genome-wide CRISPRa library and whole-genome RNA-seq of clinical specimens from patients with differential responses to OXA treatment, we identified PRMT3 as a potential driver in OXA resistance, followed by functional validation in vitro and in vivo. However, we also identified additional candidate genes that may regulate OXA resistance (Fig. 1c; Supplementary Tables 1 and 2). Some of these genes have been implicated in playing an oncogenic role in tumorigenesis (e.g., *LUM, C1orf35, FOSL2, ST3GAL1, GLI3*) or therapeutic resistance (e.g., *HK3*)[41–46]. Interestingly, several genes (e.g., *ALDH1A3, ABCD2,* and *PRRX1*) have been implicated in the response to platinum-based chemotherapy in several cancer types[47–49]. Since therapeutic resistance could be mediated by multiple independent pathways, it will be important to perform functional validation of these candidate genes and examine their relevance in clinical samples, which may offer us additional therapeutic targets to overcome OXA resistance.

Arginine methylation regulates the biological functions of many proteins[22,50]. Our study identified IGF2BP1 as a key substrate of PRMT3, and its methylation at R452 is essential for PRMT3-mediated OXA resistance. Our proteomic analysis also identified multiple additional PRMT3-interacting proteins, which could be PRMT3 substrates. Since IGF2BP1 KD in PRMT3-OE cells did not completely abolish the effect of PRMT3-OE on cell proliferation and survival in the presence and absence of OXA treatment, suggesting the involvement of additional PRMT3 substrates in OXA resistance. This also explains why IGF2BP1 was not identified as a hit in our CRISPRa screen. Interestingly, YBX1, one of the top PRMT3-interacting proteins, has been shown to promote the expression of multidrug resistance genes and the expression of PD-L1, which could lead to therapeutic resistance in tumors[51,52]. Thus, future studies are needed to determine whether YBX1 is a bona fide substrate of PRMT3 and whether its methylation by PRMT3 plays a role in OXA resistance.

IGF2BP1, which belongs to the IGF2BP RNA-binding protein family that includes IGF2BP1, IGF2BP2, and IGF2BP3, has been shown to regulate multiple oncogenic processes through stabilization of the mRNA of key oncoproteins (e.g., MYC)[53]. Recent studies showed that IGF2BP1 serves as a reader of m6A modification[28]. Through integrative analyses of RIP and MeRIP-m6A, and transcriptome analyses, we identified *BTBD7* and *HEG1* as two IGF2BP1 target transcripts that contain m6A modifications. Of note, both *BTBD7* and *HEG1* were not identified as hits in our CRISPRa screen, which is likely because they act downstream of PRMT3-IGF2BP1 and only partially contribute to the OXA resistance. Interestingly, our data show that methylation at R452, which is in the KH3-4 di-domain that is indispensable for m6A recognition and binding[28], plays a critical regulatory role in the function of IGF2BP1. Thus, it is possible that the methylation of IGF2BP1-R452 is crucial for the m6A recognition and binding ability of the KH3-4 di-domain in IGF2PB1. Although we focused on the role of HEG1 as an effector for IGF2BP1 function in OXA resistance, BTBD7 may be also involved in OXA resistance as it has been implicated in chemoresistance in lung Carcinoma A549 Cells[54]. Future studies are necessary to define the role of BTBD7 as a target of IGF2BP1 and its functional role in OXA resistance. Although only two IGF2BP1 target transcripts contain m6A modifications, IGF2BP1 may still enhance the stability of other RNAs without m6A as IGF2BP1 can regulate mRNA stability through m6A-independent mechanisms such as miRNA[26]. Furthermore, other IGF2BP1 target transcripts that contain m6A modifications but were not downregulated in PRMT3 KD and IGF2BP1 KD cells may also play a role in OXA resistance due to the ability of IGF2BP1 in regulating translation[55]. Moreover, m6A RNA readers also control mRNA fate by regulating splicing, structure, decay, and subcellular localization. Further studies are necessary to define the scope of IGF2BP1's m6A-dependent and -independent functions as well as its role in the regulation of other aspects of mRNA biology other than stability.

Our results indicated that high PRMT3 expression levels strongly correlate with poor clinical outcomes and therapeutic responses to

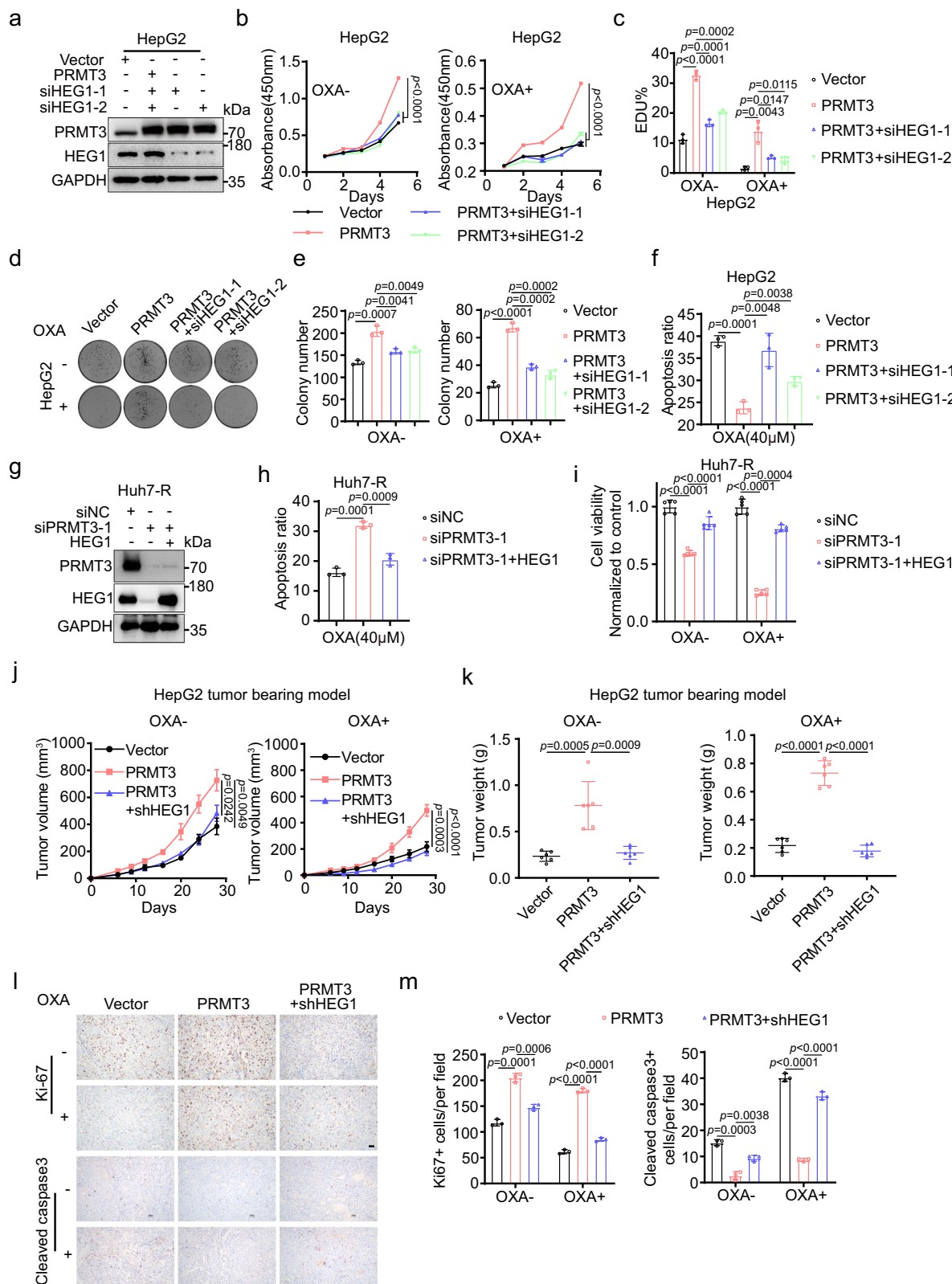

OXA-based HAIC in HCC patients. Therefore, clinically, PRMT3 expression in biopsy specimens may help to guide individualized therapeutic strategies for patients with advanced HCC before treatment. Patients with higher PRMT3 expression may need more intensive treatment (e.g., combining OXA-based HAIC with immunotherapy or targeted therapy). However, there were limitations in our study. For example, there was no replicate in our CRISPRa screening which is less

robust than those conducted in replicates and some of the enriched gRNAs may have occurred by chance. other hits identified from our screen will be validated by functional studies or be further filtered by a second CRISPR screening. Moreover, the small sample sizes of our cohorts limited our ability to firmly establish PRMT3 as a biomarker for OXA response. Also, we only tested the response of two PDC lines to OXA and OXA + SGC707. Therefore, the potential of PRMT3 as a

**Fig. 6 | The effect of PRMT3 and IGF2BP1 on OXA resistance is dependent on HEG1. a** The efficient KD of HEG1 with two independent siRNAs was confirmed by Western blot analysis. **b** The effect of HEG1 KD on the proliferation (1 μM OXA) of HepG2 cells overexpressing PRMT3 as shown by CCK8 assay. **c** The effect of HEG1 KD on the proliferation (0.5 μM OXA) of HepG2 cells overexpressing PRMT3 as shown by EDU incorporation assay. **d**, **e** The effect of HEG1 KD on the proliferation (1 μM OXA) of HepG2 cells overexpressing PRMT3 as shown by colony formation assay. **f** The effect of HEG1 KD on the apoptosis (40 μM OXA) of HepG2 cells overexpressing PRMT3 as shown by flow cytometry analysis. **g** The efficient OE of HEG1 in PRMT3 knockdown Huh7-R cells was confirmed by Western blot analysis. **h** The effect of HEG1 OE on the apoptosis (40 μM OXA) of PRMT3 KD Huh7-R cells as shown by flow cytometry analysis. **i** The effect of HEG1 OE on the proliferation (1 μM OXA) of PRMT3 KD Huh7-R cells. **j** The effect of HEG1 KD on the tumor growth of HepG2 cells overexpressing PRMT3 treated with OXA (40 μM) or vehicle as shown by measurement of tumor volume (n = 6). **k** The effect of HEG1 KD on the tumor growth of HepG2 cells overexpressing PRMT3 treated with OXA (40 μM) or vehicle as shown by tumor weights at the endpoint of the experiment on day 25 (n = 6). **l**, **m** IHC staining for Ki67 and cleaved Caspase 3 in tumors from subQ implanted HepG2 cells, PRMT3-OE HepG2 cells, and PRMT3-OE/HEG1 KD HepG2 cells treated with OXA or vehicle. Scale bars=100 μm. For **b**–**f**, **h**, **l** and **m**, n = 3 biologically independent samples. For **i**, n = 5 biologically independent samples. For **a**, **g** and **l**, n = 3 independent experiments. Data in **b**, **c**, **e**, **f**, **h**, **i**, **j**, **k** and **m** are presented as mean ± SD and were analyzed by two-sided Student's t test. The one-way ANOVA adjusted was used for multiple comparisons for **j**. Source data are provided as a Source Data file.

biomarker for OXA response should be investigated in larger cohorts of patients and more PDC lines in the future. Also, since OXA-based HAIC combined OXA with 5-Fu, it will be interesting to determine whether PRMT3 also play a role in the response of HCC to 5-Fu.

## Methods

### Ethics statement, patients and tissues
This work was approved by the Ethics Committee of Sun Yat-Sen University Cancer Center (Ethics Approval ID: GZR2021-175). Tissue samples for screening were prospectively obtained from HCC patients who received HAIC at the Sun Yat-sen University Cancer Center, Guangzhou, China, from 2020 to 2021. Samples were divided into Response and Non-Response groups after HAIC treatment evaluated by mRECIST criterion. Thirty-six tissue samples for efficacy prediction were retrospectively obtained from HCC patients who received HAIC followed by surgical resection at the Sun Yat-sen University Cancer Center from 2015 to 2018. From July 16 2020 until June 2 2021, we enrolled 32 patients diagnosed with advanced HCC at the Sun Yat-sen University Cancer Center for the prospective study. Tissue samples were prospectively obtained from HCC patients who received HAIC through needle biopsy. Written informed consent was obtained from each patient. The study design conformed to the ethical guidelines of the 1975 Declaration of Helsinki.

### Cell lines and cell culture
Human HCC cell lines, PLC-8024 (JNO-206), Huh7 (JNO-22049) and HepG2 (JNO-10-14-3), were purchased from the Guangzhou jenniobio Biotechnology with STR (short tandem repeat) appraisal certificates. All cell lines were tested negative for mycoplasma contamination. Cells were maintained in Dulbecco's Modified Eagle medium (DMEM; Thermo Fisher, USA) supplemented with 10% fetal bovine serum (FBS; Gibco, California, USA) at 37 °C in 5% CO2.

### Genome-wide CRISPR activation screen
In this study, the CRISPR-Pool™ SAM human library (Addgene, cat. no. 1000000074) was used to identify genes responsible for OXA resistance in HCC cells. The workflow of this forward genetic screen is illustrated in Supplementary Fig. 1a. To established a stable dCas9-expressing HCC cell line (HepG2-dCas9), lentiviral containing dCas9 coding sequence was used to infect HepG2 cells with polybrene (6.0 μg/ml, GeneCopoeia, Rockville, USA). After 72 h of transduction, HepG2 cells were subjected to 5 μg/mL blasticidin (Gibco; California, USA) selection for several days. Then we transduced HepG2-Cas9 with CRISPR-Pool™ SAM human library which contains 70,290 unique sgRNA sequences targeting 23,430 human genes at a low MOI (-0.3) to ensure effective barcoding of individual cells. Then, the transduced cells were selected with 1.6 μg/ml of puromycin for 7 days to generate a mutant cell pool, which was then treated with vehicle and OXA (2 μM) for 7 days, respectively. After treatment, at least $3 \times 10^7$ cells were collected for genomic DNA extraction (Gentra Puregene Cell kit, QIAGEN #158388) to ensure over 400× coverage of CRISPR-Pool™ SAM human library. The sgRNA sequences were amplified using NEBNext® High-Fidelity 2X PCR Master Mix and subjected to massive parallel amplicon sequencing carried out by Novogene Technology (Beijing, China). The sgRNA read count and hits calling were analyzed by MAGeCK v0.5.7 algorithm. Briefly, the read counts of each sgRNA from different samples were normalized to adjust for the effect of library sizes and read count distributions. Resistant genes are afterward identified by looking for genes whose sgRNAs are ranked consistently higher using robust rank aggregation (RRA). Genes with smaller RRA values ranked higher in the activation screening. To confirm the efficient activation of the targets genes by CRISPRa system, we transfected plasmids containing sgRNAs for the target gene used in a previous study[24] into HepG2 cells and found that the expressions of all target genes including *PRMT3* were significantly upregulated (Supplementary Fig. 1h).

### Cell proliferation, colony formation, and apoptosis assays
For the cell proliferation assay, 1000 cells were seeded into 96-well plates, cell viability was assessed for 5 consecutive days with or without oxaliplatin treatment (1μM) by the Cell Counting Kit-8 (CCK8) (Dojindo, Japan). For the colony formation assay, 1000 cells were seeded into 6-well plates for about 10 days with or without oxaliplatin treatment (0.5 μM), which were stained with crystal violet and counted at the endpoint. All studies were conducted in triplicates. For the apoptosis assay, cells were treated with oxaliplatin (40 μM) for 48 h. The cells were labeled with Annexin V/APC and 7-AAD (KeyGEN Bio-TECH, China) according to the manufacturer's instructions, the gating strategy for apoptosis assay measured by flow cytometry was shown in Supplementary Fig. 9.

### Immunoblotting (IB)
Cells were harvested and lysed with lysis buffer. Proteins were extracted and loaded in SDS-PAGE, and transferred onto PVDF membrane (Millipore, Billerica, MA, USA). After blocking with Bovine Serum Albumin (Beyotime, China) and sequential incubation with the primary antibody: anti-β-Actin antibody (Absin, Abs830031ss, 1:1000 dilution), anti-GAPDH antibody (Proteintech, 60004-1-Ig, 1:2000 dilution), anti-PRMT3 antibody (Abcam, Ab191562, 1:2000 dilution), anti-IGF2BP1 antibody (Proteintech, 22803-1-Ap, 1:1000 dilution), anti-ADMA antibody (Cell Signaling Technology, 13522S, 1:1000 dilution), anti-FLAG antibody (Cell Signaling Technology, #14793, 1:1000 dilution) and anti-HEG1 antibody (Bioss, bs-15449R, 1:750 dilution) and secondary antibodies: anti-mouse IgG (Cell Signaling Technology, 7076S, 1:3000 dilution) and anti-rabbit IgG (Cell Signaling Technology, 7074S, 1:3000 dilution) (Supplementary Table 1), the blots were detected using the ECL detection kit (Millipore).

### Immunoprecipitation (IP), and mass spectrometry analysis of PRMT3 interaction and IGF2BP1 arginine methylation
Cells were lysed with IP lysis buffer (Thermo Fisher Scientific) supplemented with proteinase inhibitor (Sigma-Aldrich), incubated on ice for 15 min, and cleared by centrifugation at 12,000 × g at 4 °C for 20 min. After the pre-clearing step with protein G-agarose beads (Thermo Fisher Scientific), cell lysate (2 mg) was subjected to IP with

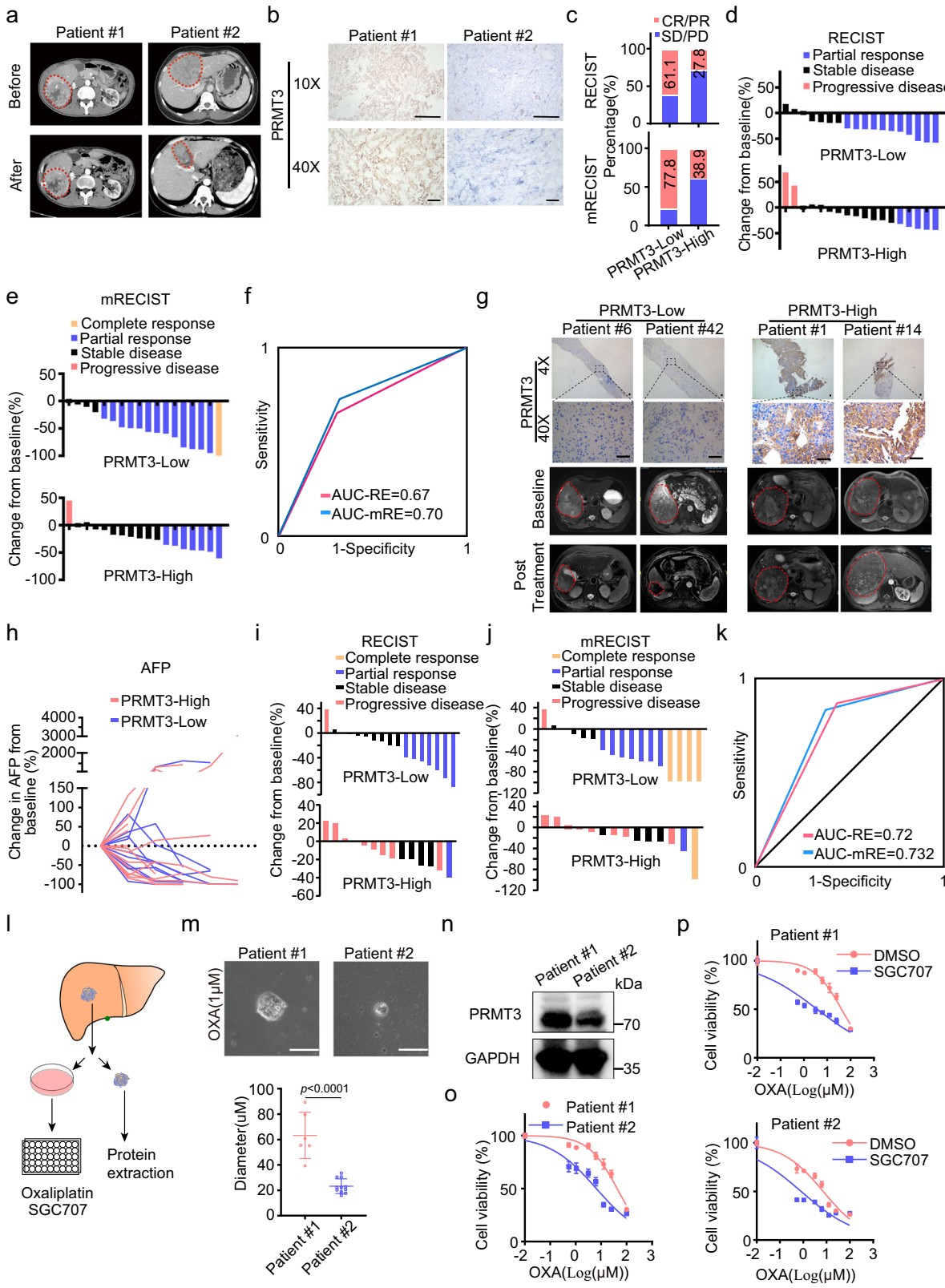

the indicated antibodies (anti-PRMT3 antibody, Abcam, Ab191562, 1:50 dilution; anti-IGF2BP1 antibody, Proteintech, 22803-1-Ap, 4 μg; anti-FLAG antibody, Cell Signaling Technology, #14793, 4 μg; Rabbit IgG, Proteintech, B900610, 2 μg) overnight at 4 °C. Then, immune complexes were washed three times in cold lysis buffer. The input and output samples were resolved by SDS-PAGE and detected by immunoblot analysis with the indicated primary antibodies: anti-

GAPDH antibody (Proteintech, 60004-1-Ig, 1:2000 dilution), anti-PRMT3 antibody (Abcam, Ab191562, 1:2000 dilution), anti-IGF2BP1 antibody (Proteintech, 22803-1-Ap, 1:1000 dilution) and anti-ADMA antibody (Cell Signaling Technology, 13522S, 1:1000 dilution). As secondary antibody, we used anti-mouse IgG (Cell Signaling Technology, 7076S, 1:3000 dilution) and anti-rabbit IgG (Cell Signaling Technology, 7074S, 1:3000 dilution) (Supplementary Table 1). The

**Fig. 7 | High PRMT3 expression correlates with poor clinical outcomes and poor therapeutic responses to OXA-based HAIC in HCC patients. a, b** MRI images of HCC with lower and higher PRMT3 expression before (upper) and after (lower) OXA-based HAIC treatment. Scale bar, 100 μm. **c** The responses of the SYSUCC HCC cohort to OXA-based HAIC were determined by RECIST and modified RECIST criteria. Waterfall plots depicting the maximum response of intrahepatic target lesions by RECIST (**d**) and modified RECIST (**e**) in the PRMT3-high ($n = 18$ patients) /low ($n = 18$ patients) groups in the SYSUCC HCC cohort treated with OXA-based HAIC. **f** AUC curves for predicting the responses of HCC patients to OXA-based HAIC using PRMT3 expression in post-treatment tumor samples and the RECIST and modified RECIST criteria. **g** The baseline and post-treatment MRI images of HCC patients, who had low ($n = 2$ patients) and high ($n = 2$ patients) PRMT3 expression, respectively, showed the patients' response to the OXA-based HAIC. Scale bar, 50 μm. **h** The change in levels of AFP for all patients with elevated AFP at baseline ($n = 24$ patients). Waterfall plots depicting the maximum response of intrahepatic target lesions in the prospective study using the RECIST (**i**) and modified RECIST criteria (**j**) in the PRMT3-low ($n = 17$ patients) /high ($n = 14$ patients) groups. **k** AUC curves for predicting objective response to OXA-based HAIC using PRMT3 expression and the RECIST and modified RECIST criteria. **l** A scheme illustrating the process for isolating primary HCC and the generation of patient-derived cell (PDC) models. **m** The bright field images and sizes of the tumor colonies for the two PDC lines treated with OXA (1 μM) for 48 h. Patient #1 ($n = 6$), Patient #1 ($n = 11$). Scale bar = 50 μm. **n** PRMT3 expression in the two PDC lines. **o** The dose-response and IC50 values of OXA in the two PDC lines. **p** The effect of SGC707 on the sensitivity of PDC lines to OXA treatment. For **b**, **g** and **n**, $n = 3$ independent experiments. For **o** and **p**, $n = 3$ biologically independent samples. Data in **m**, **o** and **p** are presented as mean ± SD. Data were analyzed by two-sided Student's $t$ test in **m**. Source data are provided as a Source Data file.

uncropped and unprocessed scans of the blots were provided in the Source Data file.

## Immunofluorescent (IF) and immunohistochemical (IHC) staining

For IF analysis of cultured cells, HCC cells were grown on chamber slides precoated with poly (L-lysine). Cells were fixed with cold paraformaldehyde. For the analysis of HCC tissues, we used formalin-fixed paraffin-embedded tissues (FFPE). Cultured cells or paraffin sections were permeabilized with PBS containing 0.1% Triton X-100, and blocked with AquaBlock (East Coast Bio, North Berwick, ME). Cells were probed with the following primary antibodies: anti-PRMT3 antibody (Abcam, Ab191562, 1:100 dilution) and anti-IGF2BP1 antibody (Santacruz Biotechnology, Sc-166344, 1:500 dilution). After washing the cells with PBS-T three times, the cells were incubated with Alexa Fluor™ 594 goat anti-rabbit IgG (H + L) (Thermo Fisher Scientific, #R37117, 1:200 dilution) or Alexa Fluor$^R$ 488 goat anti-mouse IgG (H + L) (Thermo Fisher Scientific, #A-11008,1:200 dilution) secondary antibodies and DAPI-containing mounting solution VECTASHIELD (Vector Laboratories). The slices were visualized by using a Nikon inverted microscope Eclipse Ti-U equipped with a digital camera, or a Nikon A1 laser scanning confocal microscope at the Center for Advanced Microscopy/Nikon Imaging Center (CAM).

For IHC staining, all human tissue research in this study was conducted according to protocols approved by the Sun Yat-sen University Cancer Center, Guangzhou, China. The HCC tissue sections were stained with following antibodies: anti-PRMT3 antibody (Abcam, Ab191562, 1:100 dilution), anti-Ki67 antibody (Abcam, Ab15580, 1:1000 dilution) and anti-Cleaved-caspase 3 antibody (Cell Signaling Technology, #9664, 1:2000 dilution) as described in Supplementary Table 1.

## RNA-seq and quantitative RT real-time PCR (qPCR)

Total RNA was extracted using RNA-Quick Purification Kit (ES Science, Guangzhou, China) and cDNA synthesis using the Prime-Script cDNA synthesis kits (Invitrogen, California, USA) according to the manufacturer's instructions. The reverse-transcribed cDNA products were used for qPCR analysis using SYBR Green PCR kit (Invitrogen, California, USA). Supplementary Table 2 includes detailed information about the sequence of the used primers.

## Small interfering RNA for PRMT3, IGF2BP1, HEG1

Small interfering RNA for PRMT3, IGF2BP1, HEG1 were purchased from Genepharma (Shanghai, China). Reverse transfection of small interfering RNA was performed with Lipofectamine-RNAiMAX (Invitrogen, Carlsbad, CA). After 24 h, the supernatant was replaced with fresh medium and the downregulation efficiency was identified by qRT-PCR and western blot 48 h after cotransfections. Targeting sequences were listed in Supplementary Table 3.

## MeRIP (m6A)-seq and RNA-immunoprecipitation seq, RIP-qPCR (RIP-qPCR)

For meRIP-seq, total RNA in PLC-8024 cells was extracted by using TRIzol™ Reagent (Invitrogen™, Cat# 15596018) and detected by Bioptic Qseq100 Bio-Fragment Analyzer (Bioptic Inc.). DNase I (Invitrogen™, Cat# EN0525) treatment was adopted to remove DNA contamination. Additional phenol-chloroform isolation and ethanol precipitation treatments were performed to remove enzyme contamination. For meRIP-Seq, 20 μg purified RNA was fragmented into ~200 nucleotide-long fragments by incubating in magnesium RNA fragmentation buffer for 6 min at 70 °C. The fragmentation was stopped by adding EDTA. Then, Zymo RNA Clean and Concentrator-5 Kit was used to purify fragmented total RNA (Zymo Research™, Cat# R1013). Next, m6A immunoprecipitation was performed by using EpiTM m6A immunoprecipitation kit (Epibiotek™, Cat# R1804). Briefly, protein A magnetic beads (Invitrogen™, Cat# 10002D), protein G magnetic beads (Invitrogen™, Cat# 10004D) and anti-N6-methyladenosine (m6A) Antibody (Sigma-Aldrich™, Cat#ABE572) were mixed together and incubated at 4°C overnight. After the beads-antibody incubation, the beads were recovered by magnet and resuspended within 5X precipitation buffer solution and RNase Inhibitor and incubated at 4 °C for another 2 h. The beads-antibody-RNA mixture was washed twice with high-salt buffer and twice with low-salt IP buffer. After extensive washing, bound RNA was eluted from the beads with wash buffer solution, then additional phenol-chloroform isolation and ethanol precipitation treatment were performed to purify the bound RNA.

For RIP- Seq, cells were collected and then the pellet was resuspended in lysis buffer and rotated for 30 min at 4 °C. After cell lysis, harvested the lysate by centrifugation at 12,000 × $g$ for 10 min. Transfer the supernatant into a fresh 1.5 ml tube. Note that adding protease inhibitor and RNase inhibitor into the lysis buffer. Keep about 10% volume of lysate and exacted the RNA as the input to detecting the RNA integrity. The following RIP steps were performed by using EpiTM RNA-immunoprecipitation kit (EpibiotekTM, Cat#R1819). Forty microliters of protein G beads was washed twice with IP buffer and added into the lysate together with the antibody, followed by incubation overnight at 4°C. After incubation, transfer the supernatant into a fresh 1.5 ml tube. Recover the beads by magnet and resuspended within 1× wash buffer, rotated at 4 °C for 10 min. Remove the supernatant and repeat the washing step three times. Extracted the co-precipitated RNA by TRIzol™ Reagent (Invitrogen™, Cat# 15596018) and Phenol-chloroform method. Co-precipitated RNA and input RNA were subjected to library construction by using EpiTM mini longRNA-seq kit (Epibiotek, Cat# E1802) according to the manufacturer's protocols. Briefly, reverse transcription was performed using random primers and the ribosome cDNA (cDNA fragments originating from rRNA molecules) was removed after cDNA synthesis using probes specific to

mammalian rRNA. The directionality of the template-switching reaction not only preserves the 5′ end sequence information of RNA but also the strand orientation of the original RNA. Libraries for immunoprecipitated RNA were PCR amplified for 18 cycles. Library quality was determined using Qseq100 Bio-Fragment Analyzer (Bioptic Inc.). The strand-specific libraries were sequenced on Illumina Novaseq 6000 system with paired-end 2 × 150 bp read length.

## Animal experiments

All animal experiments were approved by the Institutional Animal Care and Use Committee of Sun Yat-Sen University Cancer Center (Ethics Approval ID: L102012021003X). BALB/C nude mice were kept in an animal room with a 12-h light–dark cycle at a temperature of 20–22 °C with 40–70% humidity. For the subcutaneous tumor models, $5 \times 10^6$ PLC-8024 or HepG2 cells were injected into 4-week-old male BALB/C nude mice, and the tumor tissues were taken out 1 month later for future experiments. For the drug-resistant subcutaneous tumor models, PLC-8024 or HepG2 cells were injected subcutaneously into 4-week-old male BALB/C nude mice. Drug administration was adopted when the tumors reached about 50 mm³ in size, at which point mice were randomized for treatment with DMSO (intraperitoneally), SGC707 (20 mg/kg/every 3 days, intraperitoneally), or oxaliplatin (5 mg/kg/every 3 days, intraperitoneally). Mice were euthanized by $CO_2$ asphyxiation for tumor harvesting after the appearance of tumors with a diameter greater than 1.5 cm in any group. The weights of the excised tumors were recorded.

## Statistics and reproducibility

Statistical analysis was performed using GraphPad Prism version 8.0 for Windows. For comparing two groups, the two-tailed Student t-test was used unless otherwise stated. All boxplots indicate median (center), 25th and 75th percentiles (bounds of box), and minimum and maximum (whiskers). Experiments were performed a minimum of three times. $P < 0.05$ was considered statistically significant. All grouped data are presented as mean ± SD unless otherwise stated.

## Reporting summary

Further information on research design is available in the Nature Portfolio Reporting Summary linked to this article.

# Data availability

The sequence data generated in this study have been deposited in the GEO database under the accession number GSE206500, GSE206501, GSE206502, GSE206503, and GSE206504. The remaining data are available within the Article, Supplementary Information or Source Data file. Source data are provided with this paper.

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

## Acknowledgements

This work was supported by grants from the National Natural Science Foundation of China (No. 82172815 to B.L., 81902473 to Y.Y., 81972301 to Y.Y. and 82103601 to Y.N.). The authors would like to thank Dr. Traver Hart from the Department of Bioinformatics and Computational Biology, University of Texas MD Anderson Cancer Center for helpful advice on the CRISPR screen in the manuscript.

## Author contributions

Y.S., Y.N., and Y.Y. designed experimental approaches, performed experiments, analyzed data, and co-wrote the manuscript; Kai Li, C.Z., Z.Q., Keren Li, Z.L., Z.Y., D.Z., J.Q., W.H., C.W., and Y.L. performed experiments, analyzed data, and provided critical input; G.W., Y.Y. and B.L. designed experimental approach, analyzed data, provided oversight and critical expertise, and co-wrote the manuscript.

## Competing interests

The authors declare no competing interests.
