## [Peer Review File · Nature Communications]

Reviewers' Comments:

Reviewer #1:

Remarks to the Author:

In this manuscript, the authors described performing a CRISPRa assay on an HCC cell line, with and without oxaliplatin and found enrichment of gRNAs targeting PMRT3, which they also describe as upregulated in tumors from nonresponders to oxo. They found that in their clinic PMRT3 expression levels correlate to resistance to TACE. The overall impression is that the CRISPR screen was poorly analyzed and gRNAs targeting their gene of interest were not actually enriched on a gene-level analysis based on the data shown. There was some interesting analysis of the molecular pathways in this manuscript, particularly regarding methylation. However, the results presented here are unlikely to impact on patient care as clinical treatment with TACE is unlikely to be stratified in the way they suggest here, with biopsies and IHC of PMRT3, as patients tended to respond to the therapy regardless of the PMRT3 levels.

Major issues:

In figure 1C a volcano plot is shown of enriched and depleted gRNAs. A black dot with "note" is shown but it is not well explained what this is. Is this a single gRNA targeting PMRT3 that is enriched? What about the other gRNAs targeting this gene – are they also enriched? All of the gRNAs targeting this gene should be shown. A per-gene analysis must also be performed such as with MAGEK. A lot of analysis of the screen performance is missing such as analysis of controls – are known oncogenic drivers enriched in the screen? What happens to the non-targeting control gRNAs? Finally, no validation of the CRISPRa activity of gRNAs increasing target gene expression level is shown.

The RNA-seq data needs more analysis. Oxo is used in trans-arterial chemo-embolization (TACE), but, to my knowledge, much of the effect of TACE is thought to stem from hypoxia due to cutting off the blood supply, rather than an effect of the local chemotherapy. The RNA-seq needs gene set analysis to examine the patterns of genes linked to progression. I suspect the gene sets might include hypoxia-related genes.

In fig 2D the data indicate overexpression of PMRT3 made the cell lines more SENSITIVE rather than resistant to OXO as written in the text. This is very confusing.

Reviewer #2:

Remarks to the Author:

Shi and coworkers explored a novel molecular mechanism for the oxaliplatin resistance in hepatocellular carcinoma (HCC). By a large suite of molecular biology and genetics experiments in vitro and in vivo, the authors demonstrated that PRMT3-IGF2BP1-HEG1 axis to be a key pathway for oxaliplatin resistance: They showed IGF2BP1 is a novel target of PRMT3, methylated at R452; the methylation event is crucial for the function of IGF2BP1 in stabilizing the HEG1mRNA, which resulted oxaliplatin resistance in HCC.

The experiments are well and extensively designed and contain meaningful controls. The data are solid, presented clearly, and support the conclusions. The writing is well organized. This is an important piece of work, bringing about several novel findings spanning PRMT biology, post-transcriptional regulation, and drug resistance in oncology. It is worth of timely publication at a prestigious journal.

Minors:

Page 5, line 66: remove "the symmetric or" because PRMT3 is a type I enzyme.

Is Figure 2B correct? The figure suggests that IC50 of OXA is lower in the presence of PRMT3 expression, which is opposed to what the authors' statement on line 123-124, page 7.

It would be interesting if the authors can reveal how R452 methylation of IGF2B1 affects its interaction with m6-mRNA of HEG1.

Reviewer #3:

Remarks to the Author:

The article titled "PRMT3-mediated arginine methylation of IGF2BP1 promotes oxaliplatin resistance in liver cancer" is a logical and thorough attempt at identifying and validating potential mechanisms of oxaliplatin resistance in hepatocellular carcinoma (HCC). The authors use a screening method based on CRISPR/Cas9 activating library as well as clinical HCC samples to identify a novel PRMT3- IGF2BP1-HEG1 axis as a potential driver of oxaliplatin resistance, and subsequently employ multiple in vitro and in vivo models of HCC to validate the proposed mechanism. The authors measure IC50 and apoptosis due to oxaliplatin treatment as endpoints to verify their hypotheses at each step. The use of clinically relevant HCC samples to screen for and validate the identified axis of resistance adds to the significance of this study. However, the proposed use of PRMT3 as 'a biomarker for oxaliplatin resistance in HCC patients' or therapeutic target to improve sensitivity to oxaliplatin is not entirely convincing. I have the following questions/comments.

Major edits:

Line 46-48: "However, most HCC patients do not respond to OXA due to primary resistance. Also, the benefits in patients who respond to OXA are short-lived due to the development of acquired resistance."

- It would be advisable to cite relevant articles to support these statements to strengthen the significance of the findings from this study.

Line 95-98: "Next, the cell pools treated with vehicle or OXA for 3 days were subjected to next-generation sequencing of genomic DNA (gDNA) to identify genes that are negatively and positively associated with OXA-resistance."

&

Supplementary Line 40-42: "Then, the transduced cells were selected with 1.6 µg/ml of puromycin for 7 days to generate a mutant cell pool, which was then treated with vehicle and OXA for 7 days, respectively."

- It is not clear what is the dose of oxaliplatin that was used during the screening. Furthermore, was the duration of dosing 3 or 7 days? Was there a reasoning, such as using known Cmax values, behind using the selected dose of oxaliplatin; especially because the authors use different doses- 25/40/50 µM to check oxaliplatin sensitivity in the following experiments.

Line 121-122: "To determine the role of PRMT3 in OXA-resistance, we first examined the effect of PRMT3 overexpression (OE) on the response of HepG2 cells to OXA (Fig. 2A)."

-The authors predominantly use PRMT3 OE based cell lines as models to interrogate the role of PRMT3 in oxaliplatin resistance. Did the authors consider derivatizing oxaliplatin resistant cell lines from parental cell lines to check the expression and protein levels of PRMT3 in the derived cell lines? How do the authors rule out that the observed sensitivity of oxaliplatin after intervention with PRMT3 KO or PRMT3 inhibitor is likely due to an induced higher level of PRMT3 in the control? Can the authors justify that PRMT3 upregulation might be a transcriptional change associated with oxaliplatin treatment and not a driver of oxaliplatin resistance?

Line 122-124: "We found that PRMT3 OE significantly increased the IC50 of OXA in HepG2 cells compared to vector control (Fig. 2B)."

- The legend in Figure 2B needs to be rectified.

Line 133-134: "We found that PRMT3 KO significantly reduced the IC50 of OXA in PLC-8024 and Huh7 (Fig. 2G & Suppl. Figure 2D)."

- The authors can consider rephrasing 'significantly reduced' when showing IC50 data to mentioning the fold-change in IC50, essentially because it is not possible to derive significance (p-values) without performing triplicate IC50 experiments.

Line 164-167: "We found that PRMT3 overexpression increases the asymmetric demethylated arginine (aDMA) of IGF2BP1 in HepG2 cells (Fig. 3F). Also, PRMT3 KD or PRMT3 inhibitor SGC707 significantly reduced the aDMA of IGF2BP1 (Fig. 3F-G & Suppl. Fig. 3C), suggesting IGF2BP1 is a substrate of PRMT3."

- For figures 3F, 3G, it will be essential to quantify the changes in aDMA by measuring IP levels normalized to input levels of aDMA to better support the notion that PRMT3 OE affects aDMA levels.

Line 244-246: "Also, we found that PRMT3 or IGF2BP1 positively regulated HEG1 expression:

PRMT3 KO or IGF2BP1 KD downregulated HEG1 mRNA and protein expression level; PRMT3 OE upregulated HEG1 mRNA and protein (Fig. 5D-H).”

- Since transcriptomic changes in PRMT3 were linked to regulation of HEG1 mRNA levels, did the authors find HEG1 or IGF2BP1 during the preliminary screen in Figure 1?

Others- A stratification of TCGA datasets on liver cancer based on PRMT3 expression does not show significant difference in disease-free survival or progression-free interval between patients. Would the authors like to comment on how they suggest using 'PRMT3 expression in biopsy specimens could predict patients' response to OXA-based HAIC' using the demonstrated experiments on patient samples?

Minor edits:

Rectify spelling errors- in title 'mediated'; line 94-PLC-8024; figure 4D-vector.

In its current state, this manuscript requires major revisions and is not suitable for Nature Communications.

Reviewer Comments to the Author:

Reviewer #1 (Comments to the Author):

In this manuscript, the authors described performing a CRISPRa assay on an HCC cell line, with and without oxaliplatin and found enrichment of gRNAs targeting PMRT3, which they also describe as upregulated in tumors from nonresponders to oxo. They found that in their clinic PMRT3 expression levels correlate to resistance to TACE. The overall impression is that the CRISPR screen was poorly analyzed and gRNAs targeting their gene of interest were not actually enriched on a gene-level analysis based on the data shown. There was some interesting analysis of the molecular pathways in this manuscript, particularly regarding methylation. However, the results presented here are unlikely to impact on patient care as clinical treatment with TACE is unlikely to be stratified in the way p to respond to the therapy regards of the PMRT3 levels.

We thank this reviewer's constructive comments and the recognition of the novel mechanistic insights we offered into the development of OXA-resistance in HCC. Here we have provided detailed information on the data analysis of CRISPRa screen using MAGeCK in the Materials and Methods (**Page 23, line 479-line 484**). We have also provided additional information to show the enrichment of PRMT3 sgRNAs in our results (**Page 6, line 115-line 116**).

This reviewer's comment on the lack of significant impact of our studies on patient care is due to the misunderstanding of our study design. We have clarified the differences between trans-arterial chemo-embolization (TACE) and hepatic arterial infusion chemotherapy (HAIC) in this revised manuscript to avoid any confusion (**Page 3, line 37-line 40, also see below**) as follows:

"Chemotherapy, delivered via trans-arterial chemoembolization (TACE), which led to hypoxia, or hepatic arterial infusion (HAIC), which does not lead to hypoxia, has been used as a major therapeutic approach in the treatment of HCC."

We have addressed this reviewer's major concerns regarding the CRISPRa screen as described below. We believe that the quality of our study is significantly improved by

addressing these concerns.

1. In figure 1C a volcano plot is shown of enriched and depleted gRNAs. A black dot with “note” is shown but it is not well explained what this is. Is this a single gRNA targeting PMRT3 that is enriched? What about the other gRNAs targeting this gene – are they also enriched? All of the gRNAs targeting this gene should be shown.

We thank this reviewer for the comments. The volcano plot shows the candidate genes instead of sgRNAs that were positively and negatively correlated with OXA resistance. The black dot with “note” represents “PRMT3” in **Fig. 1a-b** and we have corrected it in **Fig. 1a, b**. Yes, all three sgRNAs for PRMT3 were detected and enriched in the OXA-treated group and they were shown in **Fig. 1d**.

2. A per-gene analysis must also be performed such as with MAGEK.

We thank this reviewer for the comments. We would like to make the following clarifications: The analysis of CRISPRa screen was indeed performed with MAGECK as described in the Supplementary Material and Methods section of our previous submission (**Page 2, line 48-line 49**). We have added a brief description of this method in the Results section (**Page 5, line 95-line 97**) and in the Material and Methods section (**Page 23, line 480-line 484**) in the main text as follows:

Results: “We then used Model-based Analysis of Genome-wide CRISPR/Cas9 Knockout (MAGECK) to identify hits from our CRISPR screening as described.”

Materials and Methods: “Briefly, the read counts of each sgRNA from different samples were normalized to adjust for the effect of library sizes and read count distributions. Resistant genes were identified by looking for genes whose sgRNAs were ranked consistently higher using robust rank aggregation (RRA). Genes with smaller RRA value ranked higher in the activation screening.”

3. A lot of analysis of the screen performance is missing such as analysis of controls – are known oncogenic drivers enriched in the screen? What happens to the non-targeting control gRNAs?

We thank the reviewer for raising this insightful point. In our CRISPRa screen for genes involved in OXA-resistance, we indeed identified multiple known oncogenic drivers that have been reported in various cancer types, such as C1orf35 (myeloma),¹ FOSL2 (breast cancer, colon cancer),^{2,3} ST3GAL1 (breast cancer, ovarian cancer),^{4,5} GLI3 (colorectal cancer, breast cancer),^{6,7} SMAD2 (breast cancer, gastric cancer, pancreatic cancer),⁸⁻¹⁰ EYA2 (breast cancer, acute leukemia)^{11,12} (**see Supplementary Table 1**). Interestingly, although most of these genes have not been studied in HCC, we found that several of these genes are amplified in HCC (cBioportal), including C1orf35 (2.2%), ST3GAL1 (4%), and OSR (3%), suggesting that they may also play a role in HCC progression. We have included this information in the Discussion section (**Page 19, line 406-Page 20, line 408**).

There is no non-targeting control in this human CRISPR-Pool™ SAM CRISPRa library, which was originally developed by Dr. Feng Zhang' lab from the Broad Institute of MIT and Harvard and was deposited to Addgene (#1000000078). This library prioritizes sgRNAs with minimal off-target activity and contains a total of 70,290 guides targeting 23430 RefSeq coding isoforms (3 sgRNAs per isoform). In contrast, the human CRISPR Knockout Pooled Library (GeCKO v2) (Pooled Library #1000000048, #1000000049) that was deposited into Addgene does contain non-targeting control. This CRISPRa library has been described in detail and validated for high efficiency and specificity in the activation of gene expression in Dr. Zhang's lab,^{13,14} and has been used in multiple studies, including Zhao et al. 2019,¹⁵ Wijdeven et al. 2022,¹⁶ Joung et al. 2022,¹⁷ to name a few. Due to the lack of non-targeting control, we were not able to evaluate the noise of our screening and may have uncovered some false-positive sgRNAs as hits in our study (**Supplementary Table 1**). However, our integrative analysis of CRISPRa screen and transcriptomic profiling of clinical samples with a stringent cutoff followed by functional validation with genetic and pharmacological approaches allows us to identify PRMT3 as a driver of OXA resistance.

4. no validation of the CRISPRa activity of gRNAs increasing target gene expression

level is shown.

As suggested, we have included new data on the validation of the CRISPRa gRNA activity in the Results section (**Page 6, line 116 to line 119**) and Methods and Materials section (**Page 23, line 484 to Page 24, line 488**) as follows:

Results: “Also, we confirmed that the three PRMT3-specific sgRNAs, together with sgRNAs for the target genes used in a previous study,²³ efficiently activate their corresponding target genes when transfected into HepG2 cells (Supplementary Fig. 1e).”

Materials and Methods: “To confirm the efficient activation of the targets genes by CRISPRa system, we transfected plasmids containing sgRNAs for the target genes used in a previous study²³ into HepG2 cells and found that the expression of all target genes including PRMT3 were significantly upregulated (Supplementary Fig. 1e).”

The RNA-seq data needs more analysis. Oxo is used in trans-arterial chemoembolization (TACE), but, to my knowledge, much of the effect of TACE is thought to stem from hypoxia due to cutting off the blood supply, rather than an effect of the local chemotherapy. The RNA-seq needs gene set analysis to examine the patterns of genes linked to progression. I suspect the gene sets might include hypoxia-related genes.

As suggested, we have included Gene Set Enrichment Analysis (GSEA) of RNA-seq on OXA-responsive and non-responsive tumors (Supplementary Table 3) in the Result section (**Page 6, line 105-line 109**) (see below). We performed RNA-seq of needle biopsy samples from HCC patients treated with OXA-based chemotherapy, which is delivered via hepatic arterial infusion chemotherapy (**HAIC**), instead of trans-arterial chemoembolization (**TACE**), which indeed cut off the blood supply in the tumors. Thus, the blood supply of tumors in our patients was not cut off and there was no dramatic hypoxia due to the lack of blood supply. We have clarified the differences between TACE and HAIC in the Introduction section of this revised manuscript (**Page 3, line 37-line 40**) (see below). Consistent with the nature of HAIC, our GSEA analysis did not

identify activation of hypoxia as the top pathway in the OXA-resistant tumors (**Supplementary Table 3**). Although we did observe hypoxia in OXA-resistant tumors, it may reflect the aggressive nature of these non-responsive tumors since hypoxia is activated in many advanced cancers.

Results: “Consistent with the aggressive nature of the non-responsive tumors, Gene Set Enrichment Analysis (GSEA) showed that non-responsive tumors were enriched for the expression of genes involved in epithelial-mesenchymal-transition (EMT), IL6-JAK-STAT3, KRAS, and angiogenesis, which were among the top 10 activated pathways (Supplementary Table 3).”

Introduction: “Chemotherapy, delivered via trans-arterial chemoembolization (TACE), which led to hypoxia, or hepatic arterial infusion (HAIC), which does not lead to hypoxia, has been used as a major therapeutic approach in the treatment of HCC.”

In fig 2D the data indicate overexpression of PMRT3 made the cell lines more SENSITIVE rather than resistant to OXO as written in the text. This is very confusing. Our results indeed indicate that overexpression of PMRT3 renders the HCC cell lines more resistant to OXA as written in the text. To make it clearer, we have made changes in the wording in this revised manuscript to avoid confusion (**Page 8, line 146-line 148**) as follows:

“Moreover, PRMT3 OE significantly reduced OXA-induced apoptosis compared to the vector control, and PRMT3 inhibition by SGC707 restored the sensitivity of PRMT3-OE HepG2 cells to OXA treatment (Fig. 2d, e)”

Reviewer 2 (Comments to the Author):

Shi and coworkers explored a novel molecular mechanism for the oxaliplatin resistance in hepatocellular carcinoma (HCC). By a large suite of molecular biology and genetics experiments in vitro and in vivo, the authors demonstrated that PRMT3-IGF2BP1-HEG1 axis to be is a key pathway for oxaliplatin resistance: They showed

IGF2BP1 is a novel target of PRMT3, methylated at R452; the methylation event is crucial for the function of IGF2BP1 in stabilizing the HEG1mRNA, which resulted oxaliplatin resistance in HCC.

The experiments are well and extensively designed and contain meaningful controls. The data are solid, presented clearly, and support the conclusions. The writing is well organized. This is an important piece of work, bringing about several novel findings spanning PRMT biology, post-transcriptional regulation, and drug resistance in oncology. It is worth of timely publication at a prestigious journal.

We thank this reviewer's positive comments regarding the novelty and significance of our study.

Page 5, line 66: remove "the symmetric or" because PRMT3 is a type I enzyme.

We appreciate the reviewer for pointing out this mistake. As suggested, we have removed "the symmetric" from the text.

Is Figure 2B correct? The figure suggests that IC50 of OXA is lower in the presence of PRMT3 expression, which is opposed to what the authors' statement on line 123-124, page 7.

We appreciate the reviewer's comment on our mistake. We mislabeled these two groups and have corrected it in the revised manuscript (**Fig. 2b**)

It would be interesting if the authors can reveal how R452 methylation of IGF2BP1 affects its interaction with m6A-mRNA of HEG1.

We thank the reviewer for raising this insightful point. We agree that this is an important issue. It was reported that KH3-4 di-domain of IGF2BP1 is indispensable for m6A recognition and binding.¹⁸ Our data show that PRMT3 methylates IGF2BP1 at R452, which is in the KH3-4 di-domain (**Fig. 3I-J**). Therefore, we speculate the methylation of IGF2BP1-R452 is crucial for the recognition of m6A by the KH3-4 di-domain, which has been included in the Discussion section (**Page 21, line 431-line 435 in this revised manuscript**) as follows:

Discussion: "Interestingly, our data show that methylation at R452, which is in the KH3-4 di-domain that is indispensable for m6A recognition and binding,²⁷ plays a critical regulatory role in the function of IGF2BP1. Thus, it is possible that the methylation of

IGF2BP1-R452 is crucial for the m6A recognition and binding ability of the KH3-4 domain in IGF2PB1.”

Reviewer #3 - m6a, epigenetics - (Remarks to the Author):

The article titled “PRMT3-mediated arginine methylation of IGF2BP1 promotes oxaliplatin resistance in liver cancer” is a logical and thorough attempt at identifying and validating potential mechanisms of oxaliplatin resistance in hepatocellular carcinoma (HCC). The authors use a screening method based on CRISPR/Cas9 activating library as well as clinical HCC samples to identify a novel PRMT3- IGF2BP1- HEG1 axis as a potential driver of oxaliplatin resistance, and subsequently employ multiple in vitro and in vivo models of HCC to validate the proposed mechanism. The authors measure IC50 and apoptosis due to oxaliplatin treatment as endpoints to verify their hypotheses at each step. The use of clinically relevant HCC samples to screen for and validate the identified axis of resistance adds to the significance of this study. However, the proposed use of PRMT3 as a biomarker for oxaliplatin resistance in HCC patients’ or therapeutic target to improve sensitivity to oxaliplatin is not entirely convincing.

We thank this reviewer for the positive comments on the significance of our study and the thoughtful suggestions that helped improve the manuscript. As suggested, we have included new data using OXA-resistant HCC cell lines to demonstrate that i) PRMT3 is overexpressed in OXA-resistance HCC cells compared to the parental cells (**Fig. 1i, j**), ii) PRMT3-mediated arginine methylation of IGF2BP1 led to increased mRNA stability of HEG1 in these OXA-resistant HCC cells (**Fig. 2-6, Supplementary Fig. 2-6**), and iii) PRMT3-IGF2BP1-HEG1 axis is required for OXA-resistance in these OXA-resistant cells (**Fig. 2-6, Supplementary Fig. 2-6**). These new data strengthen our conclusion that PRMT3 can serve as a therapeutic target to improve the sensitivity to OXA treatment in HCC. We agree with the reviewer’s comments on the use of PRMT3 as a biomarker. In the revised manuscript, we only indicate that high PRMT3 expression level is strongly associated with poor response to OXA and it may be able to serve as a biomarker for predicting the response to OXA-based HAIC in the Results section (**Page 16, line 327-line 328; Page 17, line 350-line 352; Page 18, line 367-line 369; Page 18, line 381-line 383**) and have included the limitation of our study

(e.g., the small sample sizes) in the Discussion section (**Page 21, line 449-Page 22, line 458**) as follows:

Results:

“High PRMT3 expression correlates with poor clinical outcomes and poor therapeutic responses to OXA-based HAIC in HCC patients” (**Page 16, line 327-line 328**)

“We found that the AUCs were 0.67 and 0.70 based on RECIST and mRECIST criteria, respectively (Fig. 7f), suggesting that PRMT3 could potentially serve as a biomarker for OXA-response.” (**Page 17, line 350-line 352**)

“These data strongly suggest that PRMT3 expression levels in pre-treatment biopsy samples could also serve as a potential biomarker to stratify patients for OXA treatment.” (**Page 18, line 367-line 369**)

“Collectively, our data strongly suggest that PRMT3 may serve as a biomarker for predicting the response of HCC patients to OXA-based chemotherapy.” (**Page 18, line 381-line 383**)

Discussion: “Our results indicated that high PRMT3 expression levels strongly correlate with poor clinical outcomes and therapeutic responses to OXA-based HAIC in HCC patients. Therefore, clinically, PRMT3 expression in biopsy specimens may help to guide individualized therapeutic strategies for patients with advanced HCC before treatment. Patients with higher PRMT3 expression may need more intensive treatment (e.g., combining OXA-based HAIC with immunotherapy or targeted therapy). However, there were limitations in our study. For example, the small sample sizes of our cohorts limited our ability to firmly establish PRMT3 as a biomarker for OXA response. Also, we only tested the response of two PDC lines to OXA and OXA + SGC707. Therefore, the potential of PRMT3 as a biomarker for OXA response should be investigated in larger cohorts of patients and more PDC lines in the future.” (**Page 21, line 449-Page 22, line 458**)

Major edits:

1. Line 46-48: "However, most HCC patients do not respond to OXA due to primary resistance. Also, the benefits in patients who respond to OXA are short-lived due to the development of acquired resistance." It would be advisable to cite relevant articles to support these statements to strengthen the significance of the findings from this study.

As suggested, we have cited relevant articles to support these statements in the Introduction section (**Page 3, line 42-44 in the revised manuscript**) as follows:

"However, most HCC patients do not respond to OXA due to primary resistance.¹⁹ Also, the benefits in patients who respond to OXA are short-lived due to the development of acquired resistance.¹⁹⁻²¹"

2. Line 95-98: "Next, the cell pools treated with vehicle or OXA for 3 days were subjected to next-generation sequencing of genomic DNA (gDNA) to identify genes that are negatively and positively associated with OXA-resistance." & Supplementary Line 40-42: "Then, the transduced cells were selected with 1.6 µg/ml of puromycin for 7 days to generate a mutant cell pool, which was then treated with vehicle and OXA for 7 days, respectively." It is not clear what is the dose of oxaliplatin that was used during the screening. Furthermore, was the duration of dosing 3 or 7 days? Was there a reasoning, such as using known Cmax values, behind using the selected dose of oxaliplatin; especially because the authors use different doses- 25/40/50 µM to check oxaliplatin sensitivity in the following experiments.

We appreciate the reviewer for pointing out our mistake. The concentration of OXA we used for the screening was 2 µM and the duration was 7 days. We have corrected this in the revised manuscript (**Page 5, line 88; Page 23, line 475**). The 2 µM dose was not chosen based on Cmax but instead based on two considerations: 1) We first determined the IC50 of OXA in HepG2 cells is 4.76 µM in a 2 days viability assay (**Supplementary Figure 1c**), 2) We then test the effect of several dosages of OXA (0.5, 1, 2, 4 µM) in a 7 days treatment experiment and found that 2 µM for 7 days could significantly inhibit proliferation (95%) and induce cell death of HepG2 cells. Since we were searching for factors that are activated in HepG2 cells to promote OXA resistance, this will provide a strong selection pressure to identify potent drivers of OXA resistance. Also, we chose 0.5 µM OXA in colony formation assay to examine the oncogenic potential of a PRMT3 in promoting cancer cell growth since the assay lasted for about 10 days. Furthermore, we chose 40 µM of OXA because this concentration allows us

to efficiently distinguish the difference in the apoptosis rate between PRMT3-OE vs vector control HepG2 cells and PRMT3-KO vs PRMT3-sgControl PLC-8024 cells treated with OXA for 48 hrs. Specifically, we found that OXA with 40 μ M was efficiently to distinguish the difference in apoptosis between PRMT3-vector (about 40%) and PRMT3-OE groups (about 20%) in HepG2 cells which were plated with a 70% confluence (9×10^5 cells per well in a 6-well plate). For PLC-8024 cells, which exhibited higher IC50 (about 15~20 μ M), OXA with 40 μ M was efficiently to distinguish the difference in apoptosis between PRMT3-sgNC (about 20%) and PRMT3-KO groups or SGC707-treated group (about 40%) which were plated with a 70% confluence (7×10^5 cells per well in a 6-well plate). We have included these data in the Results section (**Page 5, line 87-line 93**) and Materials and Methods (**Page 23, line 474-line 476; Page 24, line 490-line 497**)

Results: “HepG2 cell overexpressing dCas9 protein infected with lentiviral sgRNAs were treated with vehicle or 2 μ M OXA for 7 days. We chose 2 μ M OXA based on the IC50 study and the effects of various concentrations of OXA (0.5, 1, 2, 4 μ M) on cell proliferation and cell death of HepG2 cells during the 7 days of treatment. We found that 2 μ M OXA treatment for 7 days significantly inhibited proliferation (95%) and induced cell death of HepG2 cells (Supplementary Fig. 1d), which would provide a strong selection pressure to uncover drivers of OXA-resistance in the screening” (**Page 5, line 87-line 93**).

Materials and methods:

“Then, the transduced cells were selected with 1.6 μ g/ml of puromycin for 7 days to generate a mutant cell pool, which was then treated with vehicle and OXA (2 μ M) for 7 days, respectively.” (**Page 23, line 474-line 476**)

“For the cell proliferation assay, 1000 cells were seeded into 96-well plates, cell viability was assessed for 5 consecutive days with or without oxaliplatin treatment (1 μ M) by the Cell Counting Kit-8 (CCK-8) (Dojindo, Japan). For the colony formation assay, 1000

cells were seeded into 6-well plates for about 10 days with or without oxaliplatin treatment (0.5 μ M), which were stained with crystal violet and counted at the endpoint. All studies were conducted in triplicates. For the apoptosis assay, cells were treated with oxaliplatin (40 μ M) for 48 hours. The cells were labeled with Annexin V/APC and 7-AAD (KeyGEN BioTECH, China) according to the manufacturer's instructions." (Page 24, line 490-line 497)

3. Line 121-122: "To determine the role of PRMT3 in OXA-resistance, we first examined the effect of PRMT3 overexpression (OE) on the response of HepG2 cells to OXA (Fig. 2A)."

(1) The authors predominantly use PRMT3 OE based cell lines as models to interrogate the role of PRMT3 in oxaliplatin resistance. Did the authors consider derivatizing oxaliplatin resistant cell lines from parental cell lines to check the expression and protein levels of PRMT3 in the derived cell lines?

We appreciate the reviewer's insightful comment. Now we have generated OXA-resistant (OXA-R) HCC cell lines (PLC-8024-R, Huh7-R) and examined the mRNA and protein expression of PRMT3 in OXA-resistant HCC cell lines compared to the corresponding parental lines. We found that PRMT3 expression was upregulated at the mRNA and protein levels in OXA-R cell lines and the new data have been included in the revised manuscript (Fig.1i, j). Importantly, using these OXA-resistant cell lines, we demonstrated that the PRMT3-IGF2BP1-HEG1 axis similarly operates in these cell lines to promote OXA-resistance. Specifically, we first showed that PRMT3 KD significantly sensitized OXA-R cell lines to OXA treatment (Fig. 2g-k; Supplementary Fig. 2e, f, h). In addition, we showed that PRMT3 methylated IGF2BP1 at R452, which was required for OXA resistance in OXA-R cells (Fig. 3d, e, g, 4g, h). Furthermore, we showed that PRMT3-IGF2BP1 regulated HEG1 expression in an m6A-dependent manner in OXA-R cells (Fig. 5d-g, r; Supplementary Fig. 5f, g). Moreover, we showed that HEG1 OE rescued the effects of PRMT3 KD on cell proliferation with and without OXA treatment and OXA-induced apoptosis of OXA-R cells (Fig. 6g-i). Collectively, our data suggest that PRMT3 is a driver in OXA resistance, which is in part mediated through IGF2BP1-HEG1. These new data have been included in the

Results section (**Page 7, line 123-line 132; Page 8, line 153-line 160; Page 9, line 184-Page 10, line 193; Page 11, line 214; Page 11, line 219-line 224; Page 13, line 275-Page 14, line 277, Page 14, line 286-line 289; Page 15, line 310-line 313; Page 15, line 316**) as follows:

Results:

“To examine whether PRMT3 is overexpressed in OXA-resistant HCC cell lines, we generated OXA-resistant PLC-8024 and Huh7 cells (PLC-8024-R & Huh7-R) by subjecting these cells to OXA treatment for 6 months. We then compared the IC₅₀ of OXA in the parental lines and the OXA-resistant sublines. We found that OXA-resistant sublines (PLC-8024-R and Huh7-R) have a much higher IC₅₀ (fold change >5) than their parental counterpart (Figure 1g; Supplementary Fig. 1g). Also, OXA-induced apoptosis was dramatically reduced in PLC-8024-R & Huh7-R compared to their parental controls (Figure 1h; Supplementary Fig. 1h). Importantly, PRMT3 expression is upregulated at the mRNA and protein levels in both PLC-8024-R and Huh7-R cells compared to their parental counterpart (Fig. 1i, j).” (**Page 7, line 123-line 132**)

“The efficiency of PRMT3 KD by two independent siRNAs was confirmed by Western blot (Fig. 2f). We found that PRMT3 KO/KD reduced the IC₅₀ of OXA in PLC-8024, Huh7 and the OXA-R sublines (Fig. 2g; Supplementary Fig. 2e). Also, PRMT3 KO/KD in PLC-8024, Huh7, PLC-8024-R, and Huh7-R cells potentiated OXA-mediated growth suppression as shown by colony formation assay and CCK-8 viability assay (Fig. 2h; Supplementary Fig. 2f-h). Furthermore, PRMT3 KO/KD in PLC-8024, Huh7, and Huh7-R cells sensitize these cells to OXA-induced apoptosis compared to WT cells as shown by FACS analysis of Annexin V staining (Fig. 2i; Supplementary Fig. 2i).” (**Page 8, line 153-line 160**)

“Indeed, we found that endogenous PRMT3 efficiently pulled down endogenous IGF2BP1 in PLC-8024 and Huh7-R cells (Fig. 3c, d). Similarly, endogenous IGF2BP1 efficiently pulled down PRMT3 in PLC-8024 and Huh7 cells (Fig. 3c-d). Also,

immunofluorescence (IF) staining showed that PRMT3 and IGF2BP1 colocalize in the cytoplasm in PLC-8024, Huh7, Huh7-R, and HCC tissue from patients (Fig. 3e; Supplementary Fig. 3b). We then examined the effect of PRMT3 OE on arginine methylation of IGF2BP1. We found that PRMT3 OE increased the asymmetric dimethylarginine (ADMA) of IGF2BP1 in HepG2 cells (Fig. 3f). Also, PRMT3 KD or PRMT3 inhibitor SGC707 significantly reduced the ADMA of IGF2BP1 in PRMT3-OE HepG2, PLC-8024, Huh7 and Huh7-R cells (Fig. 3f, g; Supplementary Fig. 3c), suggesting IGF2BP1 is a substrate of PRMT3.” (Page 9, line 184- Page 10, line 193)

“Additionally, IGF2BP1 KD sensitized Huh7-R cells to OXA-induced apoptosis (Fig. 4c).” (Page 11, line 214)

“The expression of IGF2BP1 WT almost fully rescued the defect in cell proliferation caused by IGF2BP1-KD and reduced OXA-induced apoptosis in PLC-8024 and Huh7-R cells (Fig. 4f-h; Supplementary Fig. 4j). On the contrary, R452K mutation significantly diminished the ability of IGF2BP1 in promoting cell proliferation and suppressing OXA-induced apoptosis in PLC-8024 and Huh7-R cells (Fig. 4f-h; Supplementary Fig. 4j).” (Page 11, line 219-line 224)

“PRMT3 KO/KD or IGF2BP1 KD downregulated HEG1 mRNA and protein expression levels in PLC-8024, Huh7, and Huh7-R cells (Fig. 5d-g; Supplementary Fig. 5d, e).” (Page 13, line 275-Page 14, line 277)

“As expected, IGF2BP1 KD significantly reduced the half-life of HEG1 mRNA in PLC-8024 and Huh7-R cells (Fig. 5i; Supplementary Fig. 5f). PRMT3 KO also reduced the stability of HEG1 mRNA (Fig. 5m; Supplementary Fig. 5g).” (Page 14, line 286-line 289)

“We also overexpressed HEG1 in PRMT3-KD Huh7-R cells (Fig. 6g). As expected, HEG1 OE rescued the effects of PRMT3 KD on cell proliferation with and without OXA

treatment and OXA-induced apoptosis (Fig. 6h, i).” (Page 15, line 310-line 313)

“HEG1 KD also sensitized Huh7-R cells to OXA-induced apoptosis (Supplementary Fig. 6f).” (Page 15, line 316)

(2) How do the authors rule out that the observed sensitivity of oxaliplatin after intervention with PRMT3 KO or PRMT3 inhibitor is likely due to an induced higher level of PRMT3 in the control?

We appreciate this reviewer’s insightful comment. We indeed observed an increase in the expression of PRMT3 in the PRMT3-WT cells treated with OXA, which indicates that PRMT3 plays a role in the adaptive response of HCC to OXA and its overexpression directly promotes resistance to HCC. To test this, we use both genetic and pharmacological approaches to examine the role of PRMT3 in the response to OXA. We showed that PRMT3-OE renders HCC cells resistant to OXA treatment (**Fig. 2b-e; Supplementary Fig. 2a**) and PRMT3-KO leads to increased sensitivity of HCC cells to OXA treatment (**Fig. 2g-q; Supplementary Fig. 2d-n**). In addition, we showed that PRMT3-specific inhibitor SGC707, which only inhibits its methyltransferase activity²² and does not influence the expression level of PRMT3 (**Fig. 3f; Supplementary Fig. 2a**), led to increased sensitivity of HCC cells to OXA (**Fig. 2j, k, o-q; Supplementary Fig. 2j, n**). Since the induction of PRMT3 expression by OXA in both the control and SGC707-treated groups are comparable (**Supplementary Fig. 2a**), our data strongly support that PRMT3 plays a direct role in the response to OXA. Collectively, our results using the PRMT3 KO or PRMT3 inhibitor demonstrated a direct role for PRMT3 in promoting OXA resistance. We have included this in the Results section (**Page 7, line 136-line 139; Page 7, line 140-Page 8, line 148; Page 8, line 154-Page 9, line 172**) as follows:

Results:

“We found that PRMT3 OE increased the IC50 of OXA in HepG2 cells compared to the vector control (Fig. 2b). Also, PRMT3 OE enhanced the growth of HepG2 cells in the

presence of OXA (Fig. 2c).” (Page 7, line 136-line 139)

“We then verified the effects of SGC707, a PRMT3-specific inhibitor,²⁴ on the response of HCC cells to OXA treatment. We found that the induction of PRMT3 expression by OXA is comparable in both the control and SGC707-treated cells (Supplementary Fig. 2a), consistent with previous reports that SGC707 inhibits PRMT3 enzymatic activities and has no effect on its protein stability.²⁴ Strikingly, SGC707 treatment completely abolished the effect of PRMT3 OE on the growth of HepG2 cells (Fig. 2c; Supplementary Fig. 2b). Moreover, PRMT3 OE significantly reduced OXA-induced apoptosis compared to the vector control, and PRMT3 inhibition by SGC707 restored the sensitivity of PRMT3-OE HepG2 cells to OXA treatment (Fig. 2d, e).” (Page 7, line 140-Page 8, line 148)

“We found that PRMT3 KO/KD reduced the IC₅₀ of OXA in PLC-8024, Huh7 and the OXA-R sublines (Fig. 2g; Supplementary Fig. 2e). Also, PRMT3 KO/KD in PLC-8024, Huh7, PLC-8024-R, and Huh7-R cells potentiated OXA-mediated growth suppression as shown by colony formation assay and CCK-8 viability assay (Fig. 2h; Supplementary Fig. 2f-h). Furthermore, PRMT3 KO/KD in PLC-8024, Huh7, and Huh7-R cells sensitize these cells to OXA-induced apoptosis compared to WT cells as shown by FACS analysis of Annexin V staining (Fig. 2i; Supplementary Fig. 2i). Moreover, PRMT3 inhibitor SGC707 treatment enhanced the OXA-induced growth suppression (Fig. 2j) and apoptosis (Fig. 2k; Supplementary Fig. 2j) in PLC-8024, Huh7, and Huh7-R cells. Interestingly, PRMT3 KO also impaired the proliferation of PLC-8024 cells in the absence of OXA treatment (Supplementary Fig. 2k), suggesting a role for PRMT3 in HCC progression. To examine the effect of PRMT3 KO on the response of HCC cells to OXA *in vivo*, we subjected tumor-bearing mice implanted with PRMT3-KO and -WT PLC-8024 cells to vehicle or OXA treatment (Supplementary Fig. 2l). We found that PRMT3 KO dramatically sensitized PLC-8024 cells to OXA treatment, as shown by reduced tumor sizes and weights (Fig. 2l-n). Similarly, SGC707 treatment improved the response of tumor-bearing mice to OXA treatment (Fig. 2o-q). Decreased expression of Ki67 and increased cleaved caspase 3 were observed in the tumor

tissues from the PRMT3-KO cells treated with OXA and tumors from PRMT3-WT cells treated with SGC707 + OXA compared to their corresponding control (Supplementary Fig. 2m-n).” (Page 8, line 154-Page 9, line 172)

(3) Can the authors justify that PRMT3 upregulation might be a transcriptional change associated with oxaliplatin treatment and not a driver of oxaliplatin resistance?

We appreciate the reviewer’s question. Yes, as shown in **Fig. 1f and Supplementary Fig. 1f**, PRMT3 upregulation is a transcriptional change associated with OXA treatment. However, we have used both gain of function and loss of function approaches to demonstrate a casual effect for PRMT3 upregulation in OXA resistance instead of just being an association with OXA treatment (also see above). Firstly, PRMT3 was identified as a candidate gene that is sufficient to promote OXA-resistance in a gain-of-function screening using CRISPR/Cas9 activation (CRISPRa) library and was found to be overexpressed in OXA-resistant HCC from patients’ samples using RNA-seq (**Fig. 1a-e**). Second, functional validation using lentiviral mediated PRMT3 OE (PRMT3), PRMT3-KO, and PRMT3-specific inhibitor treatment in multiple HCC cells in vitro and in vivo clearly demonstrates that PRMT3 is required for the OXA-resistance (**Fig. 2; Supplementary Fig. 2**). Collectively, we demonstrated that PRMT3 is indeed a driver for OXA-resistance.

4. Line 122-124: “We found that PRMT3 OE significantly increased the IC50 of OXA in HepG2 cells compared to vector control (Fig. 2B).” The legend in Figure 2B needs to be rectified.

We appreciate the reviewer for pointing out this mistake and have now corrected the legend in Fig. 2b.

5. Line 133-134: “We found that PRMT3 KO significantly reduced the IC50 of OXA in PLC-8024 and Huh7 (Fig. 2G & Suppl. Figure 2D).” The authors can consider rephrasing ‘significantly reduced’ when showing IC50 data to mentioning the fold-change in IC50, essentially because it is not possible to derive significance (p-values) without performing triplicate IC50 experiments.

As suggested, we have deleted the word “significantly” in the revised manuscript.

6. Line 164-167: “We found that PRMT3 overexpression increases the asymmetric demethylated arginine (aDMA) of IGF2BP1 in HepG2 cells (Fig. 3F). Also, PRMT3 KD or PRMT3 inhibitor SGC707 significantly reduced the aDMA of IGF2BP1 (Fig. 3F-G & Suppl. Fig. 3C), suggesting IGF2BP1 is a substrate of PRMT3.” For figures 3F, 3G, it will be essential to quantify the changes in aDMA by measuring IP levels normalized to input levels of aDMA to better support the notion that PRMT3 OE affects aDMA levels.

We thank this reviewer for the comments. We would like to make the following clarifications:

1) In our experiments as described, we measured specifically the effect of PRMT3 OE, KD, and inhibition on the ADMA of IGF2BP1. Since there is no antibody that specifically recognizes the R452me2 of IGF2BP1, we performed the immunoprecipitation of IGF2BP1 from these cells to evaluate the ADMA level of IGF2BP1 protein. Since PRMT3 OE, KD, and inhibition doesn't affect the protein expression of IGF2BP1 as shown by an equal level of IGF2BP1 in the input and the IP samples, the changes in ADMA signal will reflect the effect of PRMT3 on IGF2BP1. This approach is well accepted and widely used to identify substrates of protein-arginine methyltransferases and evaluate the changes in ADMA level.²³⁻²⁶

2) We expect that many proteins in the input lysate contain asymmetric di-methylation of arginine as shown by PhosphoSitePlus database (<https://www.phosphosite.org/>), which is not able to show the ADMA level of IGF2BP1 protein using Western blot. Thus, the ADMA WB of input lysates from PRMT3 OE, KD, and inhibition and controls cells only provided information on the effect of PRMT3 on the global level of ADMA. Also, the normalization of the ADMA signal from the immunoprecipitated IGF2BP1 to the corresponding input cannot lead to the conclusion that PRMT3 regulates the arginine methylation of IGF2BP1.

Line 244-246: “Also, we found that PRMT3 or IGF2BP1 positively regulated HEG1 expression: PRMT3 KO or IGF2BP1 KD downregulated HEG1 mRNA and protein expression level; PRMT3 OE upregulated HEG1 mRNA and protein (Fig. 5D-H).” Since transcriptomic changes in PRMT3 were linked to regulation of HEG1 mRNA levels, did the authors find HEG1 or IGF2BP1 during the preliminary screen in Figure 1?

We thank the reviewer for this insightful comment. No, we did not identify HEG1 or IGF2BP1 in my CRISPR/Cas9 screening as shown in Figure 1. IGF2BP1 is one of the many PRMT3 substrates, and HEG1 is one of the many IGF2BP1-regulated genes, both of which function downstream of PRMT3 and only partially account for the PRMT3-mediated OXA-resistant phenotype (**Figure 4a, 6c-m**). We believe that this might be the main reason why IGF2BP1 and HEG1 were not identified as a candidate driver during the preliminary screen. We have included this in our Discussion (**Page 20, line 414-line 419; Page 21, line 429-line 431**) as follows:

Discussion:

“Our proteomic analysis also identified multiple additional PRMT3-interacting proteins, which could be PRMT3 substrates. Since IGF2BP1 KD in PRMT3-OE cells did not completely abolish the effect of PRMT3-OE on cell proliferation and survival in the presence and absence of OXA treatment, suggesting the involvement of additional PRMT3 substrates in OXA-resistance. This also explains why IGF2BP1 was not identified as a hit in our CRISPRa screen.” (**Page 20, line 414-line 419**)

“Of note, both BTBD7 and HEG1 were not identified as hits in our CRISPRa screen, which is likely because they act downstream of PRMT3-IGF2BP1 and only partially contribute to the OXA resistance.” (**Page 21, line 429-line 431**)

Others- A stratification of TCGA datasets on liver cancer based on PRMT3 expression does not show significant difference in disease-free survival or progression-free interval between patients. Would the authors like to comment on how they suggest using ‘PRMT3 expression in biopsy specimens could predict patients’ response to OXA-based HAIC’ using the demonstrated experiments on patient samples?

Fig. R1 Kaplan-Meier analysis for PRMT3 expression in the TCGA cohort.

We thank the reviewer for raising this insightful point. First, we have analyzed the TCGA PanCancer Atlas Liver Hepatocellular Carcinoma RNA-seq data, which has 363 Hepatocellular Carcinoma tumor samples with mRNA expression data. We separated these tumor samples into PRMT3-high vs PRMT3-low based on the medium expression level of PRMT3 mRNA and found that PRMT3-high patients have a shorter progression-free survival and overall survival than PRMT3-low patients (**Fig. R1**), suggesting that PRMT3 may play a role in HCC progression. This notion is supported by our data showing that PRMT3-OE, -KD, and -KO modulated HCC proliferation in the absence of OXA treatment: PRMT3 OE promotes HCC proliferation in vitro and tumor growth in vivo (**Fig. 2c, 6j-m; Supplementary Fig. 2b**); PRMT3 KO suppresses HCC proliferation in vitro and tumor growth in vivo (**Fig. 2h, I-n; Supplementary Fig. 2f, g, k, m**). Also, TCGA-PanCancer HCC cohort has only 1 patient with Oxaliplatin treatment history, which cannot be used to compare the patients' responses to OXA and their survival.

In order to establish the correlation of PRMT3 overexpression with OXA response in HCC patients, we included two dependent cohorts which received oxaliplatin-based HAIC. As shown in Fig 7, patients with high PRMT3 expression exhibited a worse response to oxaliplatin-based HAIC when evaluated with both RECIST and mRECIST criteria. We also employed the ROC curves to evaluate the predictive efficacy. These results indicated that PRMT3 expression levels in pre-treatment biopsy samples are strongly associated with the responses in patients treated with oxaliplatin. We have elaborated these points in our Results section (**Page 16, line 327-line 328; Page 17, line 350-line 352; Page 18, line 367-line 369; Page 18, line 381-line 383**) and the Discussion section (**Page 21, line 449-Page 22, line 458**) as follows:

Results:

“High PRMT3 expression correlates with poor clinical outcomes and poor therapeutic responses to OXA-based HAIC in HCC patients” (**Page 16, line 327-line 328**)

“We found that the AUCs were 0.67 and 0.70 based on RECIST and mRECIST criteria, respectively (Fig. 7f), suggesting that PRMT3 could potentially serve as a biomarker for OXA-response.” (Page 17, line 350-line 352)

“These data strongly suggest that PRMT3 expression level in pre-treatment biopsy samples could also serve as a potential biomarker to stratify patients for OXA treatment.” (Page 18, line 367-line 369)

“Collectively, our data strongly suggest that PRMT3 may serve as a biomarker for predicting the response of HCC patients to OXA-based chemotherapy.” (Page 18, line 381-line 383).

Discussion: “Our results indicated that high PRMT3 expression levels strongly correlate with poor clinical outcomes and therapeutic responses to OXA-based HAIC in HCC patients. Therefore, clinically, PRMT3 expression in biopsy specimens may help to guide individualized therapeutic strategies for patients with advanced HCC before treatment. Patients with higher PRMT3 expression may need more intensive treatment (e.g., combining OXA-based HAIC with immunotherapy or targeted therapy). However, there were limitations in our study. For example, the small sample sizes of our cohorts limited our ability to firmly establish PRMT3 as a biomarker for OXA response. Also, we only tested the response of two PDC lines to OXA and OXA + SGC707. Therefore, the potential of PRMT3 as a biomarker for OXA response should be investigated in larger cohorts of patients and more PDC lines in the future.” (Page 21, line 449-Page 22, line 458)

Minor edits:

Rectify spelling errors- in title ‘mediated’; line 94-PLC-8024; figure 4D-vector.

As suggested, we have corrected it in the revised manuscript.

Reference

- 1 Luo, S. Q. *et al.* C1orf35 contributes to tumorigenesis by activating c-MYC transcription in multiple myeloma. *Oncogene* **39**, 3354-3366, doi:10.1038/s41388-020-1222-7 (2020).
- 2 Wan, X. *et al.* FOSL2 promotes VEGF-independent angiogenesis by transcriptionally activating Wnt5a in breast cancer-associated fibroblasts. *Theranostics* **11**, 4975-4991, doi:10.7150/thno.55074 (2021).
- 3 Li, S., Fang, X. D., Wang, X. Y. & Fei, B. Y. Fos-like antigen 2 (FOSL2) promotes metastasis in colon cancer. *Experimental cell research* **373**, 57-61, doi:10.1016/j.yexcr.2018.08.016 (2018).
- 4 Wu, X. *et al.* Sialyltransferase ST3GAL1 promotes cell migration, invasion, and TGF- β 1-induced EMT and confers paclitaxel resistance in ovarian cancer. *Cell death & disease* **9**, 1102, doi:10.1038/s41419-018-1101-0 (2018).
- 5 Yeo, H. L. *et al.* Sialylation of vasorin by ST3Gal1 facilitates TGF- β 1-mediated tumor angiogenesis and progression. *International journal of cancer* **144**, 1996-2007, doi:10.1002/ijc.31891 (2019).
- 6 Shen, M., Zhang, Z. & Wang, P. GLI3 Promotes Invasion and Predicts Poor Prognosis in Colorectal Cancer. *BioMed research international* **2021**, 8889986, doi:10.1155/2021/8889986 (2021).
- 7 Peng, J. & Zhang, D. Coexpression of EphA10 and Gli3 promotes breast cancer cell proliferation, invasion and migration. *Journal of investigative medicine : the official publication of the American Federation for Clinical Research* **69**, 1215-1221, doi:10.1136/jim-2021-001836 (2021).
- 8 Zhang, L. *et al.* Creatine promotes cancer metastasis through activation of Smad2/3. *Cell metabolism* **33**, 1111-1123.e1114, doi:10.1016/j.cmet.2021.03.009 (2021).
- 9 Zhou, Z. *et al.* TRIM22 inhibits the proliferation of gastric cancer cells through the Smad2 protein. *Cell Death Discov* **7**, 234, doi:10.1038/s41420-021-00627-5 (2021).
- 10 Wei, L. *et al.* Cancer-associated fibroblasts-mediated ATF4 expression promotes malignancy and gemcitabine resistance in pancreatic cancer via the TGF- β 1/SMAD2/3 pathway and ABCC1 transactivation. *Cell death & disease* **12**, 334, doi:10.1038/s41419-021-03574-2 (2021).
- 11 Patrick, A. N. *et al.* Structure-function analyses of the human SIX1-EYA2 complex reveal insights into metastasis and BOR syndrome. *Nat Struct Mol Biol* **20**, 447-453, doi:10.1038/nsmb.2505 (2013).
- 12 Ono, R., Masuya, M., Ishii, S., Katayama, N. & Nosaka, T. Eya2, a Target Activated by Plzf, Is Critical for PLZF-RARA-Induced Leukemogenesis. *Molecular and cellular biology* **37**, doi:10.1128/mcb.00585-16 (2017).
- 13 Joung, J. *et al.* Genome-scale CRISPR-Cas9 knockout and transcriptional activation screening. *Nat Protoc* **12**, 828-863, doi:10.1038/nprot.2017.016 (2017).
- 14 Konermann, S. *et al.* Genome-scale transcriptional activation by an engineered CRISPR-Cas9 complex. *Nature* **517**, 583-588, doi:10.1038/nature14136

- (2015).
- 15 Zhao, W. S. *et al.* Genome-scale CRISPR activation screening identifies a role of ELAVL2-CDKN1A axis in paclitaxel resistance in esophageal squamous cell carcinoma. *Am J Cancer Res* **9**, 1183-1200 (2019).
 - 16 Wijdeven, R. H. *et al.* CRISPR Activation Screening Identifies VGLL3-TEAD1-RUNX1/3 as a Transcriptional Complex for PD-L1 Expression. *J Immunol* **209**, 907-915, doi:10.4049/jimmunol.2100917 (2022).
 - 17 Joung, J. *et al.* CRISPR activation screen identifies BCL-2 proteins and B3GNT2 as drivers of cancer resistance to T cell-mediated cytotoxicity. *Nat Commun* **13**, 1606, doi:10.1038/s41467-022-29205-8 (2022).
 - 18 Huang, H. *et al.* Recognition of RNA N(6)-methyladenosine by IGF2BP proteins enhances mRNA stability and translation. *Nat Cell Biol* **20**, 285-295, doi:10.1038/s41556-018-0045-z (2018).
 - 19 Li, Q. J. *et al.* Hepatic Arterial Infusion of Oxaliplatin, Fluorouracil, and Leucovorin Versus Transarterial Chemoembolization for Large Hepatocellular Carcinoma: A Randomized Phase III Trial. *J Clin Oncol*, Jco2100608, doi:10.1200/jco.21.00608 (2021).
 - 20 He, M. *et al.* Sorafenib Plus Hepatic Arterial Infusion of Oxaliplatin, Fluorouracil, and Leucovorin vs Sorafenib Alone for Hepatocellular Carcinoma With Portal Vein Invasion: A Randomized Clinical Trial. *JAMA Oncol* **5**, 953-960, doi:10.1001/jamaoncol.2019.0250 (2019).
 - 21 Lyu, N. *et al.* Hepatic arterial infusion of oxaliplatin plus fluorouracil/leucovorin vs. sorafenib for advanced hepatocellular carcinoma. *Journal of hepatology* **69**, 60-69, doi:10.1016/j.jhep.2018.02.008 (2018).
 - 22 Kaniskan, H. *et al.* A potent, selective and cell-active allosteric inhibitor of protein arginine methyltransferase 3 (PRMT3). *Angewandte Chemie (International ed. in English)* **54**, 5166-5170, doi:10.1002/anie.201412154 (2015).
 - 23 Yin, S. *et al.* PRMT5-mediated arginine methylation activates AKT kinase to govern tumorigenesis. *Nature communications* **12**, 3444, doi:10.1038/s41467-021-23833-2 (2021).
 - 24 Huang, L. *et al.* PRMT5 activates AKT via methylation to promote tumor metastasis. *Nature communications* **13**, 3955, doi:10.1038/s41467-022-31645-1 (2022).
 - 25 Chan, L. H. *et al.* PRMT6 Regulates RAS/RAF Binding and MEK/ERK-Mediated Cancer Stemness Activities in Hepatocellular Carcinoma through CRAF Methylation. *Cell reports* **25**, 690-701.e698, doi:10.1016/j.celrep.2018.09.053 (2018).
 - 26 Huang, T. *et al.* PRMT6 methylation of RCC1 regulates mitosis, tumorigenicity, and radiation response of glioblastoma stem cells. *Molecular cell* **81**, 1276-1291.e1279, doi:10.1016/j.molcel.2021.01.015 (2021).

Reviewers' Comments:

Reviewer #1:

Remarks to the Author:

This manuscript is substantially improved from the original version. Some issues still need to be addressed:

1. Although the authors used a CRISPRa library that has been reported previously, the cell line used and the analysis needs more description and the screening results need more validation than what is shown here, as it required substantial customization for this screen. More details are needed about how the HepG2-Cas9 cells were made - what materials were used for this?
2. How many independent replicates of the screen were analyzed? If it was done only once, it is not sufficiently convincing that enriched gRNAs didn't occur by chance, and it should be repeated. An analysis of independent replicates should be performed to show whether the screen results were accurate and reproducible.
3. The presentation of the CRISPRa screening result should be further clarified by showing the enrichment of the three gRNAs targeting PRMT3 relative to the input for each of the replicates in a chart.
4. Also, the authors should make all of the read counts for their data available as a supplementary data table.
5. More information is needed about the CRISPR-PRMT3 cells - were they drug-selected? Clonal or mixed? Was the genomic target confirmed to be disrupted?
6. HepG2 cells are probably a hepatoblastoma-derived cell line. This should be mentioned in the manuscript. This does not necessarily detract from the results, as hepatoblastomas are often responsive to oxaliplatin, and the results were checked in HCC cell lines.

Reviewer #3:

Remarks to the Author:

The authors of the paper "PRMT3-mediated arginine methylation of IGF2BP1 enhances oxaliplatin resistance in liver cancer" have critically examined the target hypothesis and coherently answered the previously raised questions about the publication's acceptance. Furthermore, the authors have generated OXA-resistant HCC cell lines and carried out the necessary experiments to verify their hypothesis in the OXA-resistant HCC cell lines PLC-8024-R and Huh7-R.

The authors adequately justified the oxaliplatin doses employed in the various investigations. They examined the role of PRMT3 in the response to OXA using both genetic (gain of function and loss of function) and pharmacological (PRMT3 methyltransferase inhibitor) approaches and discovered a causal effect for PRMT3 overexpression in OXA resistance. They also included two dependent clinical cohorts that received oxaliplatin-based HAIC to demonstrate that patients with high PRMT3 expression had a worse response to oxaliplatin-based HAIC when assessed using both RECIST and mRECIST criteria.

The articles referenced by the authors to back up their claim that "Most HCC patients do not respond to OXA due to primary resistance.³ Furthermore, the benefits of OXA in patients are transient due to the development of acquired resistance.³⁻⁵" does not fully articulate the significance of oxaliplatin-induced acquired resistance; instead, they demonstrate the efficacy of FOLFOX, or OXA-based combination chemotherapy regimens used for HAIC. As a result, it remains to be seen whether PRMT3 may be used as a biomarker or actionable therapeutic target in the clinic for OXA- or the more clinically relevant FOLFOX-therapy resistant cases. Furthermore, the m6A signal and its overlap with IGF2BP1 RIP in Fig 5c is not clear and needs to be clarified and magnified; it does not help that there seems to only be 1 replicate of m6A-IP and IGF2BP1 RIP. However, because the authors have clearly established PRMT3 as a target associated with, and causally affecting OXA resistance in HCC using accessible biological models, I am pleased to recommend the manuscript for publication in Nature Communications.

We really appreciate this reviewer for the critical and constructive comments on our CRISPR screening, which has significantly improved the clarity of our paper.

Reviewer #1 (Remarks to the Author):

1. Although the authors used a CRISPRa library that has been reported previously, the cell line used and the analysis needs more description and the screening results need more validation than what is shown here, as it required substantial customization for this screen. More details are needed about how the HepG2-Cas9 cells were made - what materials were used for this?

We thank this reviewer for the comments. We have added more description of the cell line used, including methods for establishing the HepG2-Cas9 cells, in the Material and Methods section (**Page 23, line 480-line 484**). Also, we included detailed description of the analysis of the CRISPRa screening results by including quality control data showing the mapping ratios for vehicle- and OXA-treated cells and the boxplot showing the sgRNA representation based on sgRNA counts (**Supplementary Fig. 1e-g**) (**Page 5, line 98- page 6, line 101**). We have added more description for in the main text as follows:

Materials and Methods: “To established a stable dCas9-expressing HCC cell line (HepG2-dCas9), lentiviral containing dCas9 coding sequence was used to infect HepG2 cells with polybrene (6.0 µg/ml, GeneCopoeia, Rockville, USA). After 72 h of transduction, HepG2 cells were subjected to 5 µg/mL blasticidin (Gibco; California, USA) selection for several days.”

Results: “The quality control measurements indicated that sequencing reads had reasonable base qualities (>25) and the percentage of mapped reads (Supplementary Fig. 1e, f). Moreover, the distributions of normalized read counts in two groups were comparable (Supplementary Fig. 1g).”

Although performing more functional validation of our screening results may uncover additional genes involved in OXA-resistance, it is out of the scope of this manuscript

given the following reasons. First, the focus of our manuscript is to identify a druggable target in OXA-resistance of HCC, a significant clinical challenge for a cancer type with poor prognosis and elucidate its mechanisms of action. Our findings would enable us to pursue translational studies in the near future by developing clinical grade PRMT3 inhibitors. Second, a multi-dimensional approach allows us to prioritize PRMT3 as a candidate for functional validation and confirmed its critical role in OXA-resistance (see below). Third, we have offered in-depth mechanistic studies to elucidate how PRMT3 drives OXA-resistance. Fourth, in addition to PRMT3, our screening also identified multiple genes (top 500 hits from CRISPRa screening & genes upregulated in OXA non-responder by at least 4 folds) that have been implicated in the resistance to platinum-based chemotherapy, which includes cisplatin, oxaliplatin, carboplatin: (1) LUM, which encodes lumican, has been observed to be overexpressed in drug-resistant cell lines¹; (2) ALOX5AP was shown to overexpressed in ovarian cancer platinum non-responder and poor survival in ovarian cancer patients²; (3) ALDH1A3. Knockdowns of ALDH1A3 and treatment with inhibitors sensitized the gastric cancer stem cells to cisplatin treatment *in vitro* and *in vivo* and testicular germ cell tumors (TGCTs)³, cancer-Initiating cells from lung adenocarcinoma⁴; (4) FOSL2. FOS signaling was shown activated with treatment of oxaliplatin in relation to control in ovarian cancer⁵; (5) ABCA1. Down-regulation of ABCA1 mediated the effect of valproic acid on cisplatin sensitivity of non-small cell lung cancer cells⁶; (6) ABCD2. knocking down ABCD2 *in vitro* led to increased apoptosis in ovarian cancer cell line SKOV3 after cisplatin treatment⁷; (7) PRRX1. PRRX1 overexpression may induce MDR including cisplatin resistance in breast cancer and lung cancer^{8,9}; (8) LYN. LYN could be induced by cisplatin treatment¹⁰. (9) TUBB3. TUBB3 mediated oxaliplatin resistance in colorectal cancer and predicted cisplatin resistance in esophageal squamous cell carcinoma^{11,12}. Thus, even though our dataset is not as robust as a screen with replicates (also see below), we were still able to identify clinically relevant genes involved in OXA-resistance. We have included this in the discussion (**Page 20, line 414- line 419**):

Discussion: “Interestingly, several genes (e.g., ALDH1A3, ABCD2, and PRRX1) have

been implicated in the response to platinum-based chemotherapy in several cancer types.⁴⁷⁻⁴⁹ Since therapeutic resistance could be mediated by multiple independent pathways, it will be important to perform functional validation of these candidate genes and examine their relevance in clinical samples, which may offer us additional therapeutic targets to overcome OXA resistance.”

2. How many independent replicates of the screen were analyzed? If it was done only once, it is not sufficiently convincing that enriched gRNAs didn't occur by chance, and it should be repeated. An analysis of independent replicates should be performed to show whether the screen results were accurate and reproducible.

We thank the reviewer for raising this insightful point. We don't have replicate in our CRISPRa screening. We agree with this reviewer that our screening without replicates is less robust than those conducted in replicates and some of the enriched gRNAs may have occurred by chance. Despite this limitation, quality control analysis showed that our screening results are of high quality: (1) our CRISPR screening has mapping ratios of 72.8% and 68.7% for vehicle and OXA-treated cells, respectively (**Supplementary Fig 1f**); (2) boxplot of sgRNA count shows that sgRNA representation are similar in vehicle- and OXA-treated cells (distributions of normalized read counts by box plots: similar read count distributions) (**Supplementary Fig 1g**).

Since we don't have independent replicates, we employed several approaches to select candidate genes for functional validation. First, we combined our screening results with genome-wide RNA-seq of clinical samples from patients defined as responders and non-responders, which allows the identification of genes that are highly expressed in HCC patients with poor response to OXA. Also, we applied a stringent cutoff ($\text{Log}_2\text{FC} \geq 4$) for the differentially expressed genes identified from our RNA-seq. Furthermore, we examined the expression of PRMT3 in clinical samples and found that higher PRMT3 expression correlates with poor response to oxaliplatin treatment, which strongly suggest that PRMT3 is a key driver for oxaliplatin resistance. Therefore, despite that our CRISPR screening on its own may not be highly robust, our RNA-seq sample from clinical data can reduce the chances that we pick up the false positive hit for our functional validation. Second, we performed our functional

validation of PRMT3 in the regulation of OXA-resistance by using multiple approaches (genetic: loss-of-function and gain-of-function; pharmacological: PRMT3 inhibitor) in multiple model systems (cell line, patient-derived cells; HCC cell lines with acquired resistance to OXA). It is possible that other hits showed in Fig. 1c may not play any role in OXA-resistance due to the lack of replicates in our screening. These issues will be addressed by performing additional replicates to filter candidate genes for functional validation in our future studies.

Furthermore, CRISPR screening strategy with no replicates was also used in other published studies to identify key regulators of ferroptosis, restriction factors of SARS-CoV-2, drivers of sorafenib resistance and PARP inhibitor resistance, regulators of cell viability, factors for AKT activating, inducers of replicative stress, factors required for hepatitis A virus infection and non-canonical LC3 lipidation, regulators of naïve pluripotency, drivers of cellular senescence, combinations of candidate latency reversing agents for targeting the latent HIV-1 reservoir, chromatin regulators and protein stability regulators.¹³⁻²⁷ These findings were published in *Nature Cell Biology*, *Nature Communications*, *Advanced Science*, *Hepatology*, *The Journal of Clinical Investigation*, *Molecular Cell*, *Cell Reports* and *Science Translational Medicine*. Therefore, combining CRISPR screening results with other approaches followed by rigorous validation can overcome the limitations of CRISPR screening without replicate.

3. The presentation of the CRISPRa screening result should be further clarified by showing the enrichment of the three gRNAs targeting PRMT3 relative to the input for each of the replicates in a chart.

Since that we performed no replicate for the screen, we only showed read counts of three gRNAs targeting PRMT3 in Fig. 1d. We will validate our screen data in future studies.

4. Also, the authors should make all of the read counts for their data available as a supplementary data table.

As suggested, we have made all the read counts from the screen data available as a **Supplementary table 4**.

5. More information is needed about the CRISPR-PRMT3 cells - were they drug-selected? Clonal or mixed? Was the genomic target confirmed to be disrupted?

The PRMT3-KO cells were established by drug selection and were used as mixed KO cells. We also confirmed that the genomic target was disrupted by the genome-wide sequencing (**Supplementary Fig 2e**). The material and methods for establishing the PRMT3-KO cells was described in the Supplementary Material and Methods section of our previous submission (**Supplementary Material and Methods; Page 2, line 33-line 41**). We have added more details in the revised manuscript (**Supplementary Material and Methods; Page 2, line 33-line 42**).

Supplementary Material and Methods: “The gRNAs for PRMT3 knockout were designed using the MIT online tool CRISPRICK²⁸. The forward and reverse primers with 20 bp target sequence and inserted into the lentiCRISPRv2GFP using BsmBI sticky ends, respectively. HEK293T cells were seeded in 10 cm plate and transfected with 10 mg lentiCRISPRv2GFP-PRMT3-KO or lentiCRISPRv2 control plasmids, 5 µg psPAX2 and 2.5 µg pVSV-G plasmids using Lipofectamine 2000 to produce lentivirus. The supernatants containing lentivirus were harvested, filtered, and used to infect HCCs 48 to 72 hours post-transfection. After 72 h of transduction, HCC cells were subjected to 2 µg/mL puromycin (Gibco; California, USA) selection for several days.”

6. HepG2 cells are probably a hepatoblastoma-derived cell line. This should be mentioned in the manuscript. This does not necessarily detract from the results, as hepatoblastomas are often responsive to oxaliplatin, and the results were checked in HCC cell lines.

We thank this reviewer for the comments. We have added annotation for HepG2 cell line in the Result section (**Page 5, line 84-line 87**). As the reviewer mentioned, HepG2 is a hepatoblastoma-derived cell line, which was widely used in studies of liver cancer²⁹⁻³¹. Furthermore, our results indeed indicated that HepG2 cells is sensitive to oxaliplatin. We also validated our results in several HCC cell lines. Thus, this would not affect our conclusion.

Results: “We performed CRISPRa screen in HepG2 cells, a hepatoblastoma-derived

cell line²⁰ that is highly sensitive to OXA as compared to HCC cell lines Huh7 and PLC-8024 (IC₅₀: HepG2 4.76 μ M; Huh7: 7.89 μ M; PLC-8024: 18.19 μ M) (Supplementary Fig. 1c)."

Reviewer #3 (Remarks to the Author):

The authors of the paper "PRMT3-mediated arginine methylation of IGF2BP1 enhances oxaliplatin resistance in liver cancer" have critically examined the target hypothesis and coherently answered the previously raised questions about the publication's acceptance. Furthermore, the authors have generated OXA-resistant HCC cell lines and carried out the necessary experiments to verify their hypothesis in the OXA-resistant HCC cell lines PLC-8024-R and Huh7-R.

The authors adequately justified the oxaliplatin doses employed in the various investigations. They examined the role of PRMT3 in the response to OXA using both genetic (gain of function and loss of function) and pharmacological (PRMT3 methyltransferase inhibitor) approaches and discovered a causal effect for PRMT3 overexpression in OXA resistance. They also included two dependent clinical cohorts that received oxaliplatin-based HAIC to demonstrate that patients with high PRMT3 expression had a worse response to oxaliplatin-based HAIC when assessed using both RECIST and mRECIST criteria.

The articles referenced by the authors to back up their claim that "Most HCC patients do not respond to OXA due to primary resistance.³ Furthermore, the benefits of OXA in patients are transient due to the development of acquired resistance.³⁻⁵" does not fully articulate the significance of oxaliplatin-induced acquired resistance; instead, they demonstrate the efficacy of FOLFOX, or OXA-based combination chemotherapy regimens used for HAIC. As a result, it remains to be seen whether PRMT3 may be used as a biomarker or actionable therapeutic target in the clinic for OXA- or the more clinically relevant FOLFOX-therapy resistant cases. Furthermore, the m6A signal and its overlap with IGF2BP1 RIP in Fig 5c is not clear and needs to be clarified and magnified; it does not help that there seems to only be 1 replicate of m6A-IP and IGF2BP1 RIP. However, because the authors have clearly established PRMT3 as a target associated with, and causally affecting OXA resistance in HCC using accessible biological models, I am pleased to recommend the manuscript for publication in Nature Communications.

We thank the reviewer for their support and helpful comments throughout the review

process. We agree with the reviewer that these studies demonstrate the efficacy of FOLFOX, or OXA-based combination chemotherapy regimens used for HAIC. However, the primary resistance to OXA may be one of causes for the poor response to these treatment in non-responders. The benefits in patients who respond to OXA-based HAIC are short-lived due to the development of acquired resistance. We also made changes in the introduction (**Page 3, line 42 to line 46**). In addition, we have magnified the quality of Fig 5C in the revised Fig. 5c.

Introduction: “However, most HCC patients do not respond to OXA-based HAIC due to primary resistance.³ Also, the benefits in patients who respond to OXA-based HAIC are short-lived due to the development of acquired resistance.³⁻⁵ Thus, a better understanding of the mechanisms underlying the resistance to OXA-based chemotherapy may improve the clinical outcome of HCC patients.”

Reference

- 1 Klejewski, A. *et al.* The significance of lumican expression in ovarian cancer drug-resistant cell lines. *Oncotarget* **8**, 74466-74478, doi:10.18632/oncotarget.20169 (2017).
- 2 Ye, X. *et al.* ALOX5AP Predicts Poor Prognosis by Enhancing M2 Macrophages Polarization and Immunosuppression in Serous Ovarian Cancer Microenvironment. *Frontiers in oncology* **11**, 675104, doi:10.3389/fonc.2021.675104 (2021).
- 3 Schmidtova, S. *et al.* Disulfiram Overcomes Cisplatin Resistance in Human Embryonal Carcinoma Cells. *Cancers* **11**, doi:10.3390/cancers11091224 (2019).
- 4 Yun, X. *et al.* Targeting USP22 Suppresses Tumorigenicity and Enhances Cisplatin Sensitivity Through ALDH1A3 Downregulation in Cancer-Initiating Cells from Lung Adenocarcinoma. *Molecular cancer research : MCR* **16**, 1161-1171, doi:10.1158/1541-7786.Mcr-18-0042 (2018).
- 5 Li, Z. *et al.* lncRNA UCA1 Mediates Resistance to Cisplatin by Regulating the miR-143/FOSL2-Signaling Pathway in Ovarian Cancer. *Molecular therapy. Nucleic acids* **17**, 92-101, doi:10.1016/j.omtn.2019.05.007 (2019).
- 6 Chen, J. H. *et al.* Valproic acid (VPA) enhances cisplatin sensitivity of non-small cell lung cancer cells via HDAC2 mediated down regulation of ABCA1. *Biological chemistry* **398**, 785-792, doi:10.1515/hsz-2016-0307 (2017).
- 7 LaCroix, B. *et al.* Integrative analyses of genetic variation, epigenetic regulation, and the transcriptome to elucidate the biology of platinum sensitivity. *BMC genomics* **15**, 292, doi:10.1186/1471-2164-15-292 (2014).
- 8 Luo, H. *et al.* Paired-related homeobox 1 overexpression promotes multidrug resistance via PTEN/PI3K/AKT signaling in MCF - 7 breast cancer cells. *Molecular medicine reports* **22**, 3183-3190, doi:10.3892/mmr.2020.11414 (2020).
- 9 Zhu, H., Sun, G., Dong, J. & Fei, L. The role of PRRX1 in the apoptosis of A549 cells induced by cisplatin. *American journal of translational research* **9**, 396-402 (2017).
- 10 Singh, R. A. & Sodhi, A. Expression and activation of lyn in macrophages treated in vitro with cisplatin: regulation by kinases, phosphatases and Ca²⁺/calmodulin. *Biochimica et biophysica acta* **1405**, 171-179, doi:10.1016/s0167-4889(98)00106-2 (1998).
- 11 Wu, Y. Z., Lin, H. Y., Zhang, Y. & Chen, W. F. miR-200b-3p mitigates oxaliplatin resistance via targeting TUBB3 in colorectal cancer. *The journal of gene medicine* **22**, e3178, doi:10.1002/jgm.3178 (2020).
- 12 Gong, L., Mao, W., Chen, Q., Jiang, Y. & Fan, Y. Analysis of SPARC and TUBB3 as predictors for prognosis in esophageal squamous cell carcinoma receiving nab-paclitaxel plus cisplatin neoadjuvant chemotherapy: a prospective study. *Cancer Chemother Pharmacol* **83**, 639-647, doi:10.1007/s00280-019-03769-7 (2019).
- 13 Zhang, H. L. *et al.* PKC β II phosphorylates ACSL4 to amplify lipid peroxidation

- to induce ferroptosis. *Nature cell biology* **24**, 88-98, doi:10.1038/s41556-021-00818-3 (2022).
- 14 Mac Kain, A. *et al.* Identification of DAXX as a restriction factor of SARS-CoV-2 through a CRISPR/Cas9 screen. *Nature communications* **13**, 2442, doi:10.1038/s41467-022-30134-9 (2022).
- 15 Pettitt, S. J. *et al.* Genome-wide and high-density CRISPR-Cas9 screens identify point mutations in PARP1 causing PARP inhibitor resistance. *Nature communications* **9**, 1849, doi:10.1038/s41467-018-03917-2 (2018).
- 16 Wei, L. *et al.* Genome-wide CRISPR/Cas9 library screening identified PHGDH as a critical driver for Sorafenib resistance in HCC. *Nature communications* **10**, 4681, doi:10.1038/s41467-019-12606-7 (2019).
- 17 Zhong, C. *et al.* S100A9 Derived from Chemoembolization-Induced Hypoxia Governs Mitochondrial Function in Hepatocellular Carcinoma Progression. *Advanced science (Weinheim, Baden-Wuerttemberg, Germany)* **9**, e2202206, doi:10.1002/adv.202202206 (2022).
- 18 Niu, Y. *et al.* Loss-of-Function Genetic Screening Identifies Aldolase A as an Essential Driver for Liver Cancer Cell Growth Under Hypoxia. *Hepatology (Baltimore, Md.)* **74**, 1461-1479, doi:10.1002/hep.31846 (2021).
- 19 Koirala, P. *et al.* LncRNA AK023948 is a positive regulator of AKT. *Nature communications* **8**, 14422, doi:10.1038/ncomms14422 (2017).
- 20 Ding, Y. *et al.* Chromatin remodeling ATPase BRG1 and PTEN are synthetic lethal in prostate cancer. *The Journal of clinical investigation* **129**, 759-773, doi:10.1172/jci123557 (2019).
- 21 Sellar, R. S. *et al.* Degradation of GSPT1 causes TP53-independent cell death in leukemia while sparing normal hematopoietic stem cells. *The Journal of clinical investigation* **132**, doi:10.1172/jci153514 (2022).
- 22 Benslimane, Y. *et al.* Genome-Wide Screens Reveal that Resveratrol Induces Replicative Stress in Human Cells. *Molecular cell* **79**, 846-856.e848, doi:10.1016/j.molcel.2020.07.010 (2020).
- 23 Kulsuptrakul, J., Wang, R., Meyers, N. L., Ott, M. & Puschnik, A. S. A genome-wide CRISPR screen identifies UFMylation and TRAMP-like complexes as host factors required for hepatitis A virus infection. *Cell reports* **34**, 108859, doi:10.1016/j.celrep.2021.108859 (2021).
- 24 Li, M. *et al.* Genome-wide CRISPR-KO Screen Uncovers mTORC1-Mediated Gsk3 Regulation in Naive Pluripotency Maintenance and Dissolution. *Cell reports* **24**, 489-502, doi:10.1016/j.celrep.2018.06.027 (2018).
- 25 Wang, W. *et al.* A genome-wide CRISPR-based screen identifies KAT7 as a driver of cellular senescence. *Science translational medicine* **13**, doi:10.1126/scitranslmed.abd2655 (2021).
- 26 Ulferts, R. *et al.* Subtractive CRISPR screen identifies the ATG16L1/vacuolar ATPase axis as required for non-canonical LC3 lipidation. *Cell reports* **37**, 109899, doi:10.1016/j.celrep.2021.109899 (2021).
- 27 Dai, W. *et al.* Genome-wide CRISPR screens identify combinations of candidate latency reversing agents for targeting the latent HIV-1 reservoir.

Science translational medicine **14**, eabh3351, doi:10.1126/scitranslmed.abh3351 (2022).

- 28 Lee, H., Chang, H. Y., Cho, S. W. & Ji, H. P. CRISPRpic: fast and precise analysis for CRISPR-induced mutations via prefixed index counting. *NAR genomics and bioinformatics* **2**, lqaa012, doi:10.1093/nargab/lqaa012 (2020).
- 29 Ma, L. *et al.* LSD1-Demethylated LINC01134 Confers Oxaliplatin Resistance Through SP1-Induced p62 Transcription in HCC. *Hepatology (Baltimore, Md.)* **74**, 3213-3234, doi:10.1002/hep.32079 (2021).
- 30 Shirasaki, T. *et al.* Leukocyte cell-derived chemotaxin 2 is an antiviral regulator acting through the proto-oncogene MET. *Nature communications* **13**, 3176, doi:10.1038/s41467-022-30879-3 (2022).
- 31 Xu, Q. *et al.* HNF4 α regulates sulfur amino acid metabolism and confers sensitivity to methionine restriction in liver cancer. *Nature communications* **11**, 3978, doi:10.1038/s41467-020-17818-w (2020).

Reviewers' Comments:

Reviewer #1:

Remarks to the Author:

On this revision and rebuttal, the authors provide rationale for using an admittedly poorly robust CRISPRa screen of a hepatoblastoma cell line, combined with transcriptomic analysis of HCC samples, to chose PRMT3 for further analysis and validation as a target for OXA-resistance in liver cancer. Given the high variance of the screen and other issues such as many of the gRNAs appear to be absent from the control group, I am not convinced that PRMT3 emerged from the CRISPRa screen by more than chance, and that the screening result might not be replicated. In short, the screen was poorly executed. Many of the genes they mentioned in the rebuttal that have previous evidence had only one gRNA that was enriched in the treatment group - the genes were probably were not linked to the phenotype on a gene-level analysis. However, the authors make a sufficient argument about the extensive validation of PRMT3 and the significance of the findings to the liver cancer patient population and scientific community. I am supportive of publication.

Reviewer #1 (Remarks to the Author):

On this revision and rebuttal, the authors provide rationale for using an admittedly poorly robust CRISPRa screen of a hepatoblastoma cell line, combined with transcriptomic analysis of HCC samples, to chose PRMT3 for further analysis and validation as a target for OXA-resistance in liver cancer. Given the high variance of the screen and other issues such as many of the gRNAs appear to be absent from the control group, I am not convinced that PRMT3 emerged from the CRISPRa screen by more than chance, and that the screening result might not be replicated. In short, the screen was poorly executed. Many of the genes they mentioned in the rebuttal that have previous evidence had only one gRNA that was enriched in the treatment group - the genes were probably were not linked to the phenotype on a gene-level analysis. However, the authors make a sufficient argument about the extensive validation of PRMT3 and the significance of the findings to the liver cancer patient population and scientific community. I am supportive of publication.

We really appreciate this reviewer for the critical and constructive comments on our CRISPR screening, which has significantly improved the clarity of our paper. We have included the limitation of our screen in the discussion.

Discussion: “there was no replicate in our CRISPRa screening which is less robust than those conducted in replicates and some of the enriched gRNAs may have occurred by chance. other hits identified from our screen will be validated by functional studies or be further filtered by a second CRISPR screening.”